# RichSpace: Enriching Text-to-Video Prompt Space via Text Embedding Interpolation

## Abstract

Text-to-video generation models have made impressive progress, but they still struggle with generating videos with complex features. This limitation often arises from the inability of the text encoder to produce accurate embeddings, which hinders the video generation model. In this work, we propose a novel approach to overcome this challenge by selecting the optimal text embedding through interpolation in the embedding space. We demonstrate that this method enables the video generation model to produce the desired videos. Additionally, we introduce a simple algorithm using perpendicular foot embeddings and cosine similarity to identify the optimal interpolation embedding. Our findings highlight the importance of accurate text embeddings and offer a pathway for improving text-to-video generation performance.

## 1 Introduction

Text-to-video models have developed rapidly in recent years, driven by the advancement of Transformer architectures (Vaswani, 2017) and diffusion models (Ho et al., 2020). Early attempts at text-to-video generation focused on scaling up Transformers, with notable works such as CogVideo (Hong et al., 2022) and Phenaki (Villegas et al., 2022), which demonstrated promising results. More recently, the appearance of DiT (Peebles & Xie, 2023), which incorporates Transformers as the backbone of Diffusion Models, has pushed the capabilities of text-to-video generation models to new heights. Models like Sora (OpenAI, 2024), MovieGen (Meta, 2024), CogVideoX (Yang et al., 2024), and Veo 2 (Google, 2024) have further showcased the potential of these approaches. Despite the impressive progress made in recent years, current state-of-the-art text-to-video generation models still face challenges in effectively following complex instructions in user-provided text prompts. For example, when users describe unusual real-world scenarios, such as "a tiger with zebra-like stripes walking on grassland," the text encoder may struggle to fully capture the intended meaning. This results in text embeddings that fail to guide the video generation model toward producing the desired output. This issue is also observed in the text-to-image generation domain, where a notable work, Stable Diffusion V3 (Esser et al., 2024), addresses this challenge by incorporating multiple text encoders to improve understanding. Although their approach, which combines embeddings from different encoders, yields effective results, it comes at a significant computational cost due to the need to compute embeddings from multiple sources.

In this work, we first study the problem that prompt space is not enough to cover all video space from a theoretical perspective. We provide an informal theorem of our theoretical findings as follows:

**Theorem 1.1** (Word Embeddings being Insufficient to Represent for All Videos, informal version of Theorem 4.9). *Let $n, d$ denote two integers, where $n$ denotes the maximum length of the sentence, and all videos are in $\mathbb{R}^d$ space. Let $V \in \mathbb{N}$ denote the vocabulary size. Let $\mathcal{U} = \{u_1, u_2, \cdots, u_V\}$ denote the word embedding space, where for $i \in [V]$, the word embedding $u_i \in \mathbb{R}^k$. Let $\delta_{\min} = \min_{i,j \in [V], i \neq j} \|u_i - u_j\|_2$ denote the minimum $\ell_2$ distance of two word embedding. Let $f : \mathbb{R}^{nk} \to \mathbb{R}^d$ denote the text-to-video generation model, which is also a mapping from sentence space (discrete space $\{u_1, \ldots, u_V\}^n$) to video space $\mathbb{R}^d$. Let $M := \max_x \|f(x)\|_2, m := \min_x \|f(x)\|_2$. Let $\epsilon = ((M^d - m^d)/V^n)^{1/d}$. Then, we can show that there is a video $y \in \mathbb{R}^d$, satisfying $m \leq \|y\|_2 \leq M$, such that for any sentence $x \in \{u_1, u_2, \cdots, u_V\}^n$, $\|f(x) - y\|_2 \geq \epsilon$.*

Additionally, we take a different approach by exploring whether we can obtain a powerful text embedding capable of guiding the video generation model through interpolation within the text em-

bedding space. Through empirical experiments, we demonstrate that by selecting the optimal text embedding, the video generation model can successfully generate the desired video. Additionally, we propose an algorithm that takes advantage of perpendicular foot embeddings and cosine similarity to capture both global and local information in order to identify the optimal embedding of interpolation text (Fig. 1 and Algorithm 1).

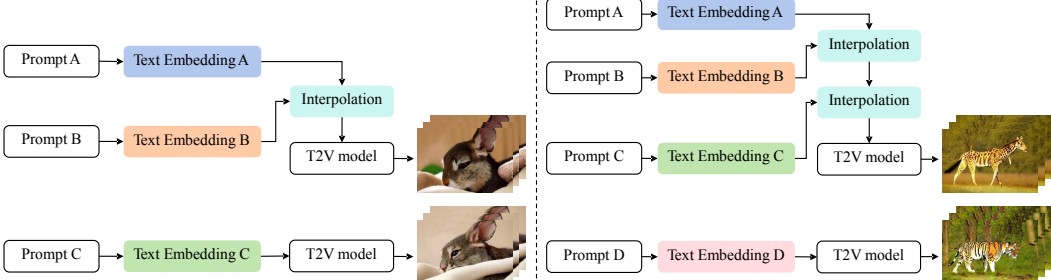

Figure 1: Two kinds of Text Prompts Mixture. **Left: Mixture of Two Prompts.** We set two prompts, A and B, and apply linear interpolation to two corresponding text embeddings. After that, we use one of the interpolation results to generate a video. To evaluate the effect of video interpolation, we set another prompt C, which describes the generated video to generate a video to compare with the interpolated video. **Right: Mixture of Three Prompts**. We set two prompts A and B and apply linear interpolation to two corresponding text embeddings. We manually choose one text embedding interpolated from A and B, then apply linear interpolation to this text embedding and text embedding C. After that, we use one of the interpolation results to generate a video. To evaluate the effect of video interpolation, we set another prompt D which describes the generated video to generate a video to compare with the interpolated video.

In summary, our main contributions are as follows:

- We demonstrate that selecting the correct text embedding can effectively guide a video generation model to produce the desired video.
- We propose a simple yet effective algorithm to find the optimal text embedding through the use of perpendicular foot embeddings and cosine similarity.

**Roadmap.** Our paper is organized as follows: Section 2 introduces our main algorithm for finding the optimal interpolation embedding. Section 3 presents the experiment result of this work. Section 4 presents the theoretical analysis, including the preliminary of our notation, key concepts of our video algorithm, model formulation, and our definition of an optimal interpolation embedding finder. In Section 5, we conclude our paper.

## 2 OUR METHODS

Section 2.1 introduces the problem formulation. In Section 2.2, we present our algorithm for finding the optimal interpolation embedding.

### 2.1 PROBLEM FORMULATION

In this section, we introduce the formal definition for finding the optimal interpolation embedding as follows:

**Definition 2.1** (Finding Optimal Interpolation Embedding Problem). *Let $P_a, P_b, P_c$ denote three text prompts. Our goal is to generate a video that contains features mentioned in $P_a$ and $P_b$, and $P_c$ is a text description of the feature combination of $P_a$ and $P_b$. Let $E_{t_a}, E_{t_b}, E_{t_c} \in \mathbb{R}^{n \times d}$ denote the text embedding of $P_a, P_b, P_c$. Let $f_\theta(E_t, z)$ be defined in Definition 4.8. We define the "Finding optimal interpolation embedding" problem as: According to $E_{t_a}, E_{t_b}, E_{t_c}$, find the optimal interpolation embedding $E_{\mathrm{opt}}$ that can make the text-to-video generation model $f_\theta(E_{\mathrm{opt}}, z)$ generate video contains features mentioned in $P_a$ and $P_b$.*

---

**Algorithm 1** Find Optimal Interpolation

---

1: **datastructure** OPTIMALINTERPFINDER
2: **members**
3:     $n \in \mathbb{N}$: the length of input sequence.
4:     $n_{\text{ids}} \in \mathbb{N}$: the ids length of input sequence.
5:     $d \in \mathbb{R}$: the hidden dimension.
6:     $E_{t_a}, E_{t_b}, E_{t_c} \in \mathbb{R}^{n \times d}$: the text embedding.
7:     $\phi_{\cos}(X, Y)$: the cosine similarity calculator.                    ▷ Definition 4.2
8: **end members**
9:
10: **procedure** OPTIMALFINDER($E_{t_a}, E_{t_b}, E_{t_c} \in \mathbb{R}^{n \times d}, n_{a_{\text{ids}}}, n_{b_{\text{ids}}}, n_{c_{\text{ids}}} \in \mathbb{N}$)
11:     /* Calculate the max ids length. */
12:     $n_{\text{ids}} \leftarrow \max\{n_{a_{\text{ids}}}, n_{b_{\text{ids}}}, n_{c_{\text{ids}}}\}$
13:     /* Truncated text embeddings. */
14:     $E_{a_{\text{truc}}} \in \mathbb{R}^{n_{\text{ids}} \times d} \leftarrow E_{t_a}[: n_{\text{ids}}, :]$
15:     $E_{b_{\text{truc}}} \in \mathbb{R}^{n_{\text{ids}} \times d} \leftarrow E_{t_b}[: n_{\text{ids}}, :]$
16:     $E_{c_{\text{truc}}} \in \mathbb{R}^{n_{\text{ids}} \times d} \leftarrow E_{t_c}[: n_{\text{ids}}, :]$
17:     /* Calculate cosine similarity, Algorithm 2. */
18:     $L_{\text{CosTruc}} \leftarrow$ COSINESIM($E_{a_{\text{truc}}}, E_{b_{\text{truc}}}, E_{c_{\text{truc}}}$)
19:     $L_{\text{CosFull}} \leftarrow$ COSINESIM($E_{t_a}, E_{t_b}, E_{t_c}$)
20:     /* Add ConsineTruc and CosineFull. */
21:     $L_{\text{CosAdd}} \leftarrow [\,]$
22:     **for** $i = 1 \rightarrow k$ **do**
23:         $L_{\text{CosAdd}}[i] \leftarrow L_{\text{CosTruc}}[i] + L_{\text{CosFull}}[i]$
24:     **end for**
25:     /* Find the optimal interpolation index. */
26:     $i_{\text{opt}} \leftarrow$ maxindex($L_{\text{CosAdd}}$)
27:     /* Calculate optimal interpolation embedding. */
28:     $E_{\text{opt}} \leftarrow \frac{i_{\text{opt}}}{k} \cdot E_{t_a} + \frac{k - i_{\text{opt}}}{k} \cdot E_{t_b}$
29:     Return $E_{\text{opt}}$
30: **end procedure**

---

We would like to refer the readers to Figure 2 (a) as an example of Definition 2.1. In Figure 2 (a), we set prompt $P_a$ to *"The tiger, moves gracefully through the forest, its fur flowing in the breeze."* and prompt $P_b$ to:*"The zebra, moves gracefully through the forest, its fur flowing in the breeze."*. Our goal is to generate a video that contains both features of "tiger" and "zebra", where we set prompt $P_c$ to *"The tiger, with black and white stripes like zebra, moves gracefully through the forest, its fur flowing in the breeze."*, to describe the mixture features of tiger and zebra. However, the text-to-video model fails to generate the expected video. Therefore, it is essential to find the optimal interpolation embedding $E_{\text{opt}}$ to make the model generate the expected video. In Figure 2 (a), the $E_{\text{opt}}$ is the 14-th interpolation embedding of $E_{t_a}$ and $E_{t_b}$.

## 2.2 OPTIMAL INTERPOLATION EMBEDDING FINDER

In this section, we introduce our main algorithm (Algorithm 3 and Algorithm 1), which is also depicted in Fig. 1. The algorithm is designed to identify the optimal interpolation embedding (as defined in Definition 2.1) and generate the corresponding video. The algorithm consists of three key steps:

1. Compute the perpendicular foot embedding (Line 9 in Algorithm 2).

2. Calculate the cosine similarity between the interpolation embeddings and the perpendicular foot embedding (Line 22 in Algorithm 2).

3. Select the optimal interpolation embedding based on the cosine similarity results (Algorithm 1).

We will now provide a detailed explanation of each part of the algorithm and the underlying intuitions.

---

**Algorithm 2** Calculate Cosine Similarity

---

1: **datastructure** COSINESIMILARITYCALCULATOR
2: **members**
3:  $n \in \mathbb{N}$: the length of input sequence.
4:  $d \in \mathbb{N}$: the hidden dimension.
5:  $E_{t_a}, E_{t_b}, E_{t_c} \in \mathbb{R}^{n \times d}$: the text embedding.
6:  $\phi_{\cos}(X, Y)$: the cosine similarity calculator.        ▷ Definition 4.2
7: **end members**
8:
9: **procedure** PERPENDICULARFOOT($E_{t_a}, E_{t_b}, E_{t_c} \in \mathbb{R}^{n \times d}$)
10:  /* Find perpendicular foot of $E_{t_c}$ on $E_{t_b} - E_{t_b}$. */
11:  $E_{ac} \leftarrow E_{t_c} - E_{t_a}$
12:  $E_{ab} \leftarrow E_{t_b} - E_{t_a}$
13:  /* Calculate the projection length. */
14:  $l_{\text{proj}} \leftarrow \langle E_{ab}, E_{ac} \rangle / \langle E_{ab}, E_{ab} \rangle$
15:  /* Calculate the projection vector. */
16:  $E_{\text{proj}} \leftarrow l_{\text{proj}} \cdot E_{ab}$
17:  /* Calculate the perpendicular foot. */
18:  $E_{\text{foot}} \leftarrow E_{t_a} + E_{\text{proj}}$
19:  Return $E_{\text{foot}}$
20: **end procedure**
21:
22: **procedure** COSINESIM($E_{t_a}, E_{t_b}, E_{t_c} \in \mathbb{R}^{n \times d}$)
23:  /* Calculate perpendicular foot. */
24:  $E_{\text{foot}} \leftarrow$ PERPENDICULARFOOT($E_{t_a}, E_{t_b}, E_{t_c}$)
25:  /* Init cosine similarity list. */
26:  $L_{\text{CosSim}} \leftarrow [\,]$
27:  **for** $i = 1 \rightarrow k$ **do**
28:   /* Compute interpolation embedding. */
29:   $E_{\text{interp}} \leftarrow \frac{i}{k} \cdot E_{t_1} + \frac{k-i}{k} \cdot E_{t_2}$
30:   /* Calculate and store cosine similarity. */
31:   $L_{\text{CosSim}}[i] \leftarrow \phi_{\cos}(E_{\text{interp}}, E_{\text{foot}})$
32:  **end for**
33:  Return $L_{\text{CosSim}}$
34: **end procedure**

---

**Perpendicular Foot Embedding.** As outlined in the problem definition (Definition 2.1), our objective is to identify the optimal interpolation embedding that allows the text-to-video generation model to generate a video containing the features described in $P_a$ and $P_b$. The combination of these features is represented by $P_c$, which typically does not lead to the desired video output. Consequently, we seek an interpolation embedding of $E_{t_a}$ and $E_{t_b}$ guided by $E_{t_c}$. The first step involves finding the perpendicular foot of $E_{t_c}$ onto the vector $E_{t_b} - E_{t_a}$, also known as the projection of $E_{t_c}$. This perpendicular foot embedding, denoted as $E_{\text{foot}}$, is not the optimal embedding in itself, as the information within $E_{t_c}$ alone does not enable the generation of the expected video. However, $E_{\text{foot}}$ serves as a useful anchor, guiding us toward the optimal interpolation embedding. Further details of this approach will be discussed in the subsequent paragraph.

**Cosine Similarity and Optimal Interpolation Embedding.** To assess the similarity of each interpolation embedding to the anchor perpendicular foot embedding $E_{\text{foot}}$, we employ the straightforward yet effective metric of cosine similarity (Definition 4.2). It is important to note that the input text prompts are padded to a fixed maximum length, $n = 266$, before being encoded by the T5 model. However, in real-world scenarios, the actual length of text prompts is typically much shorter than $n = 266$, which results in a substantial number of padding embeddings being appended to the original text prompt. The inclusion or exclusion of these padding embeddings can lead to significant differences in the perpendicular foot embedding, as their presence introduces a shift in the distribution of the text embeddings. To account for this, we treat text embeddings with and without padding separately. Specifically, we define "full text embeddings" $E_{a_t}, E_{b_t}, E_{c_t} \in \mathbb{R}^{n \times d}$ to represent the

embeddings that include padding, and "truncated text embeddings" $E_{a_{\text{truc}}}, E_{b_{\text{truc}}}, E_{c_{\text{truc}}} \in \mathbb{R}^{n_{\text{ids}} \times d}$ to represent the embeddings without padding (Line 13 in Algorithm 1). The full-text embeddings capture global information, whereas the truncated text embeddings focus on local information. We compute the perpendicular foot and cosine similarity separately for both types of text embeddings (Line 17) and then combine the results by summing the cosine similarities from the full and truncated embeddings. Finally, we select the optimal interpolation embedding based on the aggregated cosine similarity scores (Line 25).

## 3 EXPERIMENTS

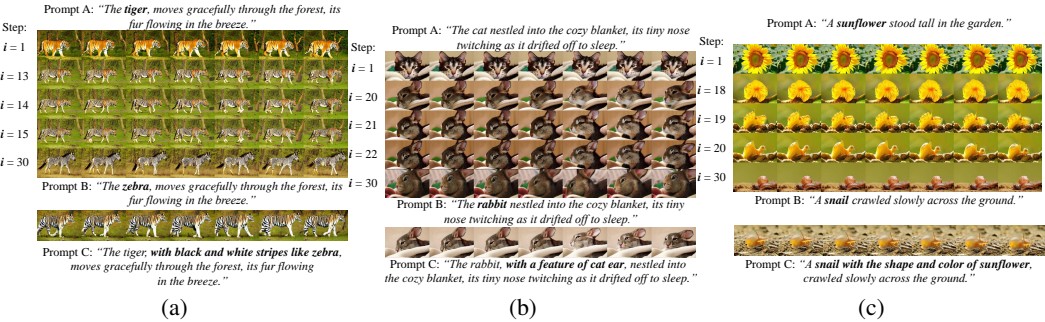

Figure 2: **Qualitative results of mixture of two features.** Figure (a): Mixture of ["Tiger"] and ["Zebra"]; Figure (b): Mixture of ["Cat"] and ["Rabbit"]; Figure (c): Mixture of ["Sunflower"] and ["Snail"]. Our objective is to mix the features described in Prompt A and Prompt B with the guidance of Prompt C. We set the total number of interpolation steps to 30. Using Algorithm 1, we identify the optimal embedding and generate the corresponding video. The video generated directly from Prompt C does not exhibit the desired mixed features from Prompts A and B.

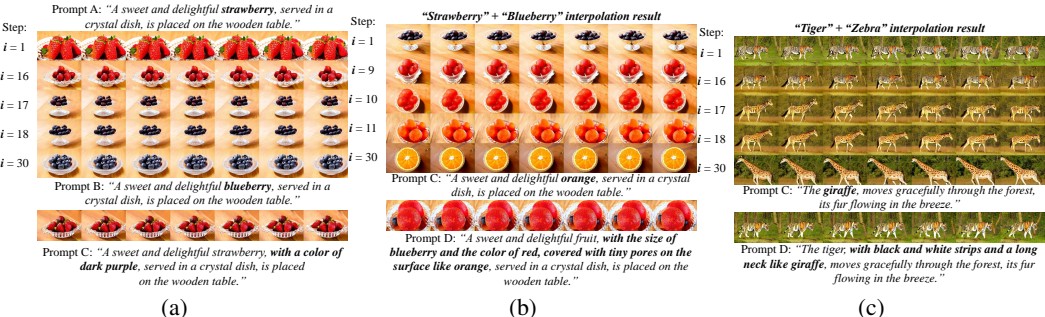

Figure 3: **Extending from two prompts mixture to three prompts mixture.** Figure (a): Mixture of ["Strawberry"] and ["Blueberry"]. Figure (b): Mixture of ["Strawberry" + "Blueberry"] and ["Orange"]. We further apply Algorithm 1 to that optimal embedding and Prompt C embedding, with the guidance of Prompt D. We identify 10-th interpolation embedding as the optimal embedding of ["Strawberry" + "Blueberry"] and ["Orange"] and generate the corresponding video. The video generated directly from Prompt D does not exhibit the desired mixed features. Figure (c): Mixture of ["Tiger" + "Zebra"] and ["Giraffe"]. We present another example of a mixture of three prompts to demonstrate the effectiveness of our algorithm.

In this section, we will first present our qualitative evaluation results of the proposed method in Section 3.1. Then, in Section 3.2, we present our quantitative evaluation.

### 3.1 QUALITATIVE EVALUATION

Our experiments are conducted on the CogVideoX-2B (Yang et al., 2024). We investigate the performance of our optimal embedding finder algorithm in the following two scenarios:

**Mixture of Features from Two Initial Prompts.** As outlined in Definition 2.1, we conduct experiments where the goal is to generate a mixture of features described in two text prompts, $P_a$ and $P_b$. We construct a third prompt, $P_c$, to specify the desired features. Following Algorithm 1, we identify the optimal text embedding and use it for the text-to-video generation with our base model. We conducted experiments using a variety of text prompts. In Figure 2 (a) and Figure 2 (b), we investigate the mixture of features from different animals, demonstrating that a video containing the mixture of tiger and zebra features, as well as the mixture of rabbit and cat features can only be generated using the optimal embedding, not directly from the text prompts. Similarly, in Figure 3 (a), we show that a video combining features from strawberry and blueberry can only be generated through the optimal embedding, highlighting a similar phenomenon in the context of fruits. Furthermore, in Figure 2 (c), we observe the same behavior in the domain of plants, specifically with the combination of rose and cactus features.

**Mixture of Features from Three Initial Prompts.** We will investigate further to see if we can add one additional feature to the video. The high-level approach involves applying our optimal interpolation embedding algorithm (Algorithm 1) twice. Given three text embeddings, $E_{t_a}$, $E_{t_b}$, and $E_{t_c}$, where we aim to blend their features in the generated video, we first apply Algorithm 1 to $E_{t_a}$ and $E_{t_b}$ to obtain the optimal interpolation embedding $E_{\text{opt}_{ab}}$. Next, we apply Algorithm 1 again, this time on $E_{\text{opt}_{ab}}$ and $E_{t_c}$, resulting in the final optimal interpolation embedding $E_{\text{opt}}$. We then use this embedding in our base model to generate the desired video. Following the method described above, we mix the giraffe feature with the tiger and zebra features, as shown in Figure 3 (c). Only by using the optimal embedding identified by our algorithm can we enable the video generation model to produce the desired video. Directly generating the video from the text prompt results in the loss of at least one of the intended features. A similar phenomenon is observed in the case of mixing strawberry, blueberry, and orange features, as shown in Figure 3 (b). The video generated directly from the text prompt always renders each object separately, failing to combine the features into a single coherent entity.

## 3.2 Quantitative Evaluation

In the previous sections, we presented the qualitative results of our method. In this section, we provide a quantitative evaluation. Following the settings used by VBench (Huang et al., 2024), we evaluate the "subject consistency" and "aesthetic quality" of the generated videos. The results for mixtures of two prompts are presented in Table 1. The average Subject Consistency (SC) of the videos generated using optimal embeddings is $0.9787$, higher than the SC of the videos generated directly from the prompt description, which is $0.9748$. As for Aesthetic Quality (AQ), the videos generated by optimal embeddings achieve a score of $0.5163$, which is lower than the $0.5519$ obtained by the videos generated from prompts.

Our method generates videos with higher "subject consistency" than those produced directly from the prompt description (i.e., Prompt C). This suggests that the optimal embedding enables the video generation model to better combine the desired features while maintaining coherence in the generated videos.

Another observation is that the "aesthetic quality" of videos generated using the optimal embeddings is lower than that of videos generated directly from text prompts. This indicates that our method better blends the desired features. The aesthetic model is trained on real-world videos, which leads to a bias toward scoring videos that resemble those found in real-world datasets. However, in our setting, we aim to expand the prompt space of the video generation model, enabling it to generate videos that are rarely observed in real-world datasets. Therefore, a lower aesthetic score reflects that our method aligns better with this goal.

## 4 Theoretical Analysis

We first introduce some basic notations in Section 4.1. In Section 4.2, we introduce formal definitions of key concepts. Then, we introduce the formal definition of each module in the CogvideoX model in Section 4.3. In Section 4.4, we provide our rigorous theoretical analysis showing that word embedding space is not sufficient to represent all videos.

Table 1: **Quantitative Evaluations.** We evaluate the videos generated using our optimal embeddings and those generated directly from the text prompt with two metrics: "Subject Consistency" (SC) and "Aesthetic Quality" (AQ). Let $f$ represent the optimal embedding finding algorithm, and $g$ denote the video generation model. A higher SC score indicates better coherence in the video, which corresponds to higher quality. Conversely, a lower AQ score suggests that the video is rarely observed in the real world, implying that it aligns more closely with the mixture of desired features. We use A to denote PromptA, B to denote PromptB and C to denote PromptC.

| Prompts | SC ($\uparrow$) | AQ ($\downarrow$) |
|---|---|---|
| ;$g(f(\text{Tiger, Zebra}))$ | **0.9751** | 0.5472 |
| $g(\text{Tiger, Zebra })$ | 0.9739 | **0.5424** |
| $g(f(\text{Cat, Rabbit}))$ | **0.9688** | **0.4649** |
| $g(\text{Cat, Rabbit})$ | 0.9608 | 0.4821 |
| $g(f(\text{Strawberry, Blueberry}))$ | **0.9920** | **0.5957** |
| $g(\text{Strawberry, Blueberry})$ | 0.9910 | 0.7256 |
| $g(f(\text{Sunflower, Snail}))$ | **0.9790** | **0.4573** |
| $g(\text{Sunflower, Snail})$ | 0.9734 | 0.4575 |
| avg. $g(f(\text{A}, \text{B}))$ | **0.9787** | **0.5163** |
| avg. $g(\text{C})$ | 0.9748 | 0.5519 |

## 4.1 NOTATIONS

For any $k \in \mathbb{N}$, let $[k]$ denote the set $\{1, 2, \cdots, k\}$. For any $n \in \mathbb{N}$, let $n$ denote the length of the input sequence of a model. For any $d \in \mathbb{N}$, let $d$ denote the hidden dimension. For any $c \in \mathbb{N}$, let $c$ denote the channel of a video. For any $n_f \in \mathbb{N}$, we use $n_f$ to denote the video frames. For any $h \in \mathbb{N}$ and $w \in \mathbb{N}$, we use $h$ and $w$ to denote the height and width of a video. For two vectors $x \in \mathbb{R}^n$ and $y \in \mathbb{R}^n$, we use $\langle x, y \rangle$ to denote the inner product between $x, y$. Namely, $\langle x, y \rangle = \sum_{i=1}^n x_i y_i$. For a vector $x \in \mathbb{R}^n$, we use $\|x\|_2$ to denote the $\ell_2$ norm of the vector $x$, i.e., $\|x\|_2 := \sqrt{\sum_{i=1}^n x_i^2}$.

Let $\mathcal{D}$ represent a given distribution. The notation $x \sim \mathcal{D}$ indicates that $x$ is a random variable drawn from the distribution $\mathcal{D}$.

## 4.2 KEY CONCEPTS

We will introduce some essential concepts in this section. We begin with introducing the formal definition of linear interpolation.

**Definition 4.1** (Linear Interpolation). *Let $x, y \in \mathbb{R}^d$ denote two vectors. Let $k \in \mathbb{N}$ denote the interpolation step. For $i \in [k]$, we define the $i$-th interpolation result $z_i \in \mathbb{R}$ as follows:*

$$z_i := \frac{i}{k} \cdot x + \frac{k-i}{k} \cdot y$$

Next, we introduce another key concept used in our paper, the simple yet effective cosine similarity calculator.

**Definition 4.2** (Cosine Similarity Calculator). *Let $X, Y \in \mathbb{R}^{n \times d}$ denote two matrices. Let $X_i, Y_i \in \mathbb{R}^d$ denote $i$-th row of $X, Y$, respectively. Then, we defined the cosine similarity calculator $\phi_{\cos}(X, Y) : \mathbb{R}^{n \times d} \times \mathbb{R}^{n \times d} \to \mathbb{R}$ as follows $\phi_{\cos}(X, Y) := \frac{1}{n} \sum_{i=1}^n \frac{\langle X_i, Y_i \rangle}{\|X_i\|_2 \|Y_i\|_2}$.*

Then, we introduce one crucial fact that we used later in this paper.

**Fact 4.3** (Volume of a Ball in $d$-dimension Space). *The volume of a $\ell_2$-ball with radius $R$ in dimension $\mathbb{R}^d$ space is $\frac{\pi^{d/2}}{(d/2)!} R^d$.*

## 4.3 MODEL FORMULATION

In this section, we will introduce the formal definition for the text-to-video generation video we use. We begin with introducing the formal definition of the attention layer as follows:

---

**Algorithm 3** Video Interpolation

---

1: **datastructure** INTERPOLATION
2: **members**
3:     $n \in \mathbb{N}$: the length of input sequence
4:     $n_f \in \mathbb{N}$: the number of frames
5:     $h \in \mathbb{N}$: the height of video
6:     $w \in \mathbb{N}$: the width of video
7:     $d \in \mathbb{N}$: the hidden dimension
8:     $c \in \mathbb{N}$: the channel of video
9:     $k \in \mathbb{N}$: the interpolation steps
10:     $T \in \mathbb{N}$: the number of inference step
11:     $E_{\mathrm{opt}} \in \mathbb{R}^{n \times d}$: the optimal interpolation embedding
12:     $E_t \in \mathbb{R}^{n \times d}$: the text embedding
13:     $f_\theta(z, E_t, t) : \mathbb{R}^{n_f \times h \times w \times c} \times \mathbb{R}^{n \times d} \times \mathbb{N} \to \mathbb{R}^{n_f \times h \times w \times c}$: the text-to-video generation model
14: **end members**
15:
16: **procedure** INTERPOLATION($E_{t_a}, E_{t_b}, E_{t_c} \in \mathbb{R}^{n \times d}, k \in \mathbb{N}, T \in \mathbb{N}$)
17:     /* Find optimal interpolation embedding, Algorithm 1. */
18:     $E_{\mathrm{opt}} \leftarrow$ OPTIMALFINDER($E_{t_a}, E_{t_b}, E_{t_c}$)
19:     /* Prepare initial latents.*/
20:     $z \sim \mathbb{N}(0, I) \in \mathbb{R}^{n_f \times h \times w \times c}$
21:     **for** $t = T \to 0$ **do**
22:         /* One denoise step. */.
23:         $z \leftarrow f_\theta(z, E_{\mathrm{opt}}, t)$
24:     **end for**
25:     Return $z$
26: **end procedure**

---

**Definition 4.4** (Attention Layer). *Let $X \in \mathbb{R}^{n \times d}$ denote the input matrix. Let $W_K, W_Q, W_V \in \mathbb{R}^{d \times d}$ denote the weighted matrices. Let $Q = XW_Q \in \mathbb{R}^{n \times d}$ and $K = XW_K \in \mathbb{R}^{n \times d}$. Let attention matrix $A = QK^\top$. Let $D := \mathrm{diag}(A\mathbf{1}_n) \in \mathbb{R}^{n \times n}$. We define attention layer* Attn *as follows:* $\mathsf{Attn}(X) := D^{-1}AXW_V$.

Then, we define the convolution layer as follows:

**Definition 4.5** (Convolution Layer). *Let $h \in \mathbb{N}$ denote the height of the input and output feature map. Let $w \in \mathbb{N}$ denote the width of the input and output feature map. Let $c_{\mathrm{in}} \in \mathbb{N}$ denote the number of channels of the input feature map. Let $c_{\mathrm{out}} \in \mathbb{N}$ denote the number of channels of the output feature map. Let $X \in \mathbb{R}^{h \times w \times c_{\mathrm{in}}}$ represent the input feature map. For $l \in [c_{\mathrm{out}}]$, we use $K^l \in \mathbb{R}^{3 \times 3 \times c_{\mathrm{in}}}$ to denote the $l$-th convolution kernel. Let $p$ denote the padding of the convolution layer. Let $s$ denote the stride of the convolution kernel. Let $Y \in \mathbb{R}^{h \times w \times c_{\mathrm{out}}}$ represent the output feature map. We define the convolution layer as follows: We use $\phi_{\mathrm{conv}}(X, c_{\mathrm{in}}, c_{\mathrm{out}}, p, s) : \mathbb{R}^{h \times w \times c_{\mathrm{in}}} \to \mathbb{R}^{h \times w \times c_{\mathrm{out}}}$ to represent the convolution operation. Let $Y = \phi_{\mathrm{conv}}(X, c_{\mathrm{in}}, c_{\mathrm{out}}, p, s)$. Then, for $i \in [h], j \in [w], l \in [c_{\mathrm{out}}]$, we have $Y_{i,j,l} := \sum_{m=1}^{3} \sum_{n=1}^{3} \sum_{c=1}^{c_{\mathrm{in}}} X_{i+m-1, j+n-1, c} \cdot K^l_{m,n,c}$*

We introduce the formal definition of linear projection layer as follows:

**Definition 4.6** (Linear Projection). *Let $X \in \mathbb{R}^{n \times d_1}$ denote the input data matrix. Let $W \in \mathbb{R}^{d_1 \times d_2}$ denote the weight matrix. We define the linear projection $\phi_{\mathrm{linear}} : \mathbb{R}^{n \times d_1} \to \mathbb{R}^{n \times d_2}$ as follows:*

$$\phi_{\mathrm{linear}}(X) := XW$$

And we define the 3D full attention layer as follows:

**Definition 4.7** (3D Attention). *Let* Attn$(X)$ *be defined as in Definition 4.4. Let $\phi_{\mathrm{conv}}(X, c_{\mathrm{in,out},p,s})$ be defined in Definition 4.5. Let $\phi_{\mathrm{linear}}(X)$ be defined as in Definition 4.6. We define the 3D attention $\phi_{\mathrm{3DAttn}}(E_t, E_v)$ containing three components:* $\phi_{\mathrm{linear}}(X)$, Attn$(X)$, $\phi_{\mathrm{conv}}(X, c_{\mathrm{in}}, c_{\mathrm{out}}, p, s)$. *Its details are provided in Algorithm 4.*

Finally, we provide the definition of the text-to-video generation model, which consists of a stack of multiple 3D attention layers, as introduced earlier.

**Definition 4.8** (Text-to-Video Generation Model). *Let $\phi_{\text{3DAttn}}$ be defined as Definition 4.7. Let $k_{\text{3D}} \in \mathbb{N}$ denote the number of 3D attention layers in the text-to-video generation model. Let $\theta$ denote the parameter in the text-to-video generation model. Let $E_t \in \mathbb{R}^{n \times d}$ denote the text embedding. Let $z \sim \mathbb{N}(0, I) \in \mathbb{R}^{n_f \times h \times w \times c}$ denote the initial random Gaussian noise. Then we defined the text-to-video generation model $f_\theta(E_t, z)$ as follows:*

$$f_\theta(E_t, z) := \underbrace{\phi_{\text{3DAttn}} \circ \cdots \circ \phi_{\text{3DAttn}}}_{k_{\text{3D}} \text{ layers}}(E_t, z).$$

### 4.4 Word Embedding Space being Insufficient to Represent for All Videos

Since the text-to-video generation model only has a finite vocabulary size, it only has finite wording embedding space. However, the space for all videos is infinite. Thus, word embedding space is insufficient to represent all videos in video space. We formalize this phenomenon to a rigorous math problem and provide our findings in the following theorem.

**Theorem 4.9** (Word Embeddings being Insufficient to Represent for All Videos, formal version of Theorem 1.1). *Let $n, d$ denote two integers, where $n$ denotes the maximum length of the sentence, and all videos are in $\mathbb{R}^d$ space. Let $V \in \mathbb{N}$ denote the vocabulary size. Let $\mathcal{U} = \{u_1, u_2, \cdots, u_V\}$ denote the word embedding space, where for $i \in [V]$, the word embedding $u_i \in \mathbb{R}^k$. Let $\delta_{\min} = \min_{i,j \in [V], i \neq j} \|u_i - u_j\|_2$ denote the minimum $\ell_2$ distance of two word embedding. Let $f : \mathbb{R}^{nk} \to \mathbb{R}^d$ denote the text-to-video generation model, which is also a mapping from sentence space (discrete space $\{u_1, \ldots, u_V\}^n$) to video space $\mathbb{R}^d$. Let $M := \max_x \|f(x)\|_2, m := \min_x \|f(x)\|_2$. Let $\epsilon = ((M^d - m^d)/V^n)^{1/d}$. Then, we can show that there exits a video $y \in \mathbb{R}^d$, satisfying $m \leq \|y\|_2 \leq M$, such that for any sentence $x \in \{u_1, u_2, \cdots, u_V\}^n$, we have $\|f(x) - y\|_2 \geq \epsilon$.*

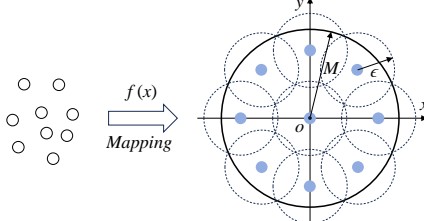

Figure 4: **Mapping from Prompt Space to Video Space.** This figure illustrates the mapping from a prompt space (with discrete prompts) to a video space (with continuous video embeddings) by a video generation model $f(x)$. Regardless of the specific form of the video generation model $f(x)$, there always exists a point in the video embedding space whose distance to all $f(x)$ is at least $\epsilon$.

Theorem 4.9 indicates that there always exists a video $y$, where its $\ell_2$ distance to all videos can be represented by the prompt embeddings is larger than $\epsilon$ (Fig. 4). This means that there always exists a video that cannot be accurately generated by using only the prompt embeddings from the word embedding space. We defer the proof to Theorem C.6 which is the restatement of Theorem 4.9 in the Appendix.

## 5 Conclusion

In this work, we propose a novel algorithm to identify the optimal text embedding, enabling a video generation model to produce videos that accurately reflect the features specified in the initial prompts. Our findings reveal that the main bottleneck in text-to-video generation is the text encoder's inability to generate precise text embeddings. By carefully selecting and interpolating text embeddings, we improve the model's ability to generate more accurate and diverse videos. From the theoretical side, we show that text embeddings generated by the text encoder are insufficient to represent all possible video features, which explains why the text encoder becomes a bottleneck in generating videos with mixed desired features. Our proposed algorithm, based on perpendicular foot embeddings and cosine similarity, provides an effective solution to these challenges. These results highlight the importance of refining text embeddings to improve model performance and lay the foundation for future advancements in text-to-video generation by emphasizing the critical role of embedding optimization in bridging the gap between textual descriptions and video synthesis.

ETHIC STATEMENT

This paper does not involve human subjects, personally identifiable data, or sensitive applications. We do not foresee direct ethical risks. We follow the ICLR Code of Ethics and affirm that all aspects of this research comply with the principles of fairness, transparency, and integrity.

REPRODUCIBILITY STATEMENT

We ensure reproducibility on both theoretical and empirical fronts. For theory, we include all formal assumptions, definitions, and complete proofs in the appendix. For experiments, we describe models and algorithms in the main text and appendix.

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

# Appendix

**Roadmap.**    In Section A, we provide a detailed discussion of our work. In Section B, we review related literature. In Section C, we provide detailed proofs for the theorem showing that word embeddings are insufficient to represent all videos. In Section D, we provide more results of our experiments. In Section E, we provide the algorithm for 3D attention.

## A    DISCUSSION

**Identifying the Actual Bottleneck of Generative Models.**    Our work identifies that the primary bottleneck hindering text-to-video generation models from producing the desired videos is the text encoder's inability to generate accurate text embeddings. Through our proposed algorithm, we can guide the video generation model to produce the desired output. This insight helps the community identify the true bottleneck within cutting-edge generative models, allowing for improvements in model performance and capabilities.

## B    RELATED WORK

**Text-to-Video Generation.**    Text-to-video generation (Singer et al., 2022; Voleti et al., 2022; Blattmann et al., 2023), as a form of conditional video generation, focuses on the synthesis of high-quality videos using text descriptions as conditioning inputs. Most recent works on video generation jointly synthesize multiple frames based on diffusion models (Song et al., 2020; Ho et al., 2020; Liu et al., 2024; Shen et al., 2024; Hu et al., 2024b;a). Diffusion models implement an iterative refinement process by learning to gradually denoise a sample from a normal distribution, which has been successfully applied to high-quality text-to-video generation. In terms of training strategies, one of the existing approaches uses pre-trained text-to-image models and inserts temporal modules (Ge et al., 2023; An et al., 2023), such as temporal convolutions and temporal attention mechanisms into the pre-trained models to build up correlations between frames in the video (Singer et al., 2022; Gu et al., 2023; Guo et al., 2023). PYoCo (Ge et al., 2023) proposed a noise prior approach and leveraged a pre-trained eDiff-I (Balaji et al., 2022) as initialization. Conversely, other works (Blattmann et al., 2023; Zhou et al., 2022a) build upon Stable Diffusion (Rombach et al., 2022) owing to the accessibility of pre-trained models. This approach aims to leverage the benefits of large-scale pre-trained text-to-image models to accelerate convergence. However, it may lead to unsatisfactory results due to the potential distribution gap between images and videos. Other approaches are training the entire model from scratch on both image and video datasets (Ho et al., 2022). Although this method can yield high-quality results, it demands tremendous computational resources.

**Enrich Prompt Space.**    In the context of conditional tasks, such as text-to-image and text-to-video models, prompts worked as conditions can have a significant influence on the performance of the models. For text-conditioned tasks, refining the user-provided natural provided natural language prompts into keyword-enriched prompts has gained increasing attention. Several recent works have explored the prompt space by the use of prompt learning, such as CoCoOp (Zhou et al., 2022b), which uses conditional prompts to improve the model's generalization capabilities. AutoPrompt (Shin et al., 2020) explores tokens with the most significant gradient changes in the label likelihood to automate the prompt generation process. Fusedream (Liu et al., 2021) manipulates the CLIP (Radford et al., 2021) latent space by using GAN (Goodfellow et al., 2014) optimization to enrich the prompt space. Specialist Diffusion (Lu et al., 2023) augments the prompts to define the same image with multiple captions that convey the same meaning to improve the generalization of the image generation network. Another work (Lin et al., 2023) proposes to generate random sentences, including source and target domain, in order to calculate a mean difference that will serve as a direction while editing. The iEdit (Bodur et al., 2024) generates target prompts by changing words in the input caption in order to retrieve pseudo-target images and guide the model. The TokenCompose (Wang et al., 2024b) and OmniControlNet (Wang et al., 2024a) control the image generation in the token-level space. Compared to the prior works, our work takes a different approach by exploring whether we can obtain a powerful text embedding capable of guiding the video generation model through interpolation within the text embedding space.

## C    WORD EMBEDDING SPACE BEING INSUFFICIENT TO REPRESENT FOR ALL VIDEOS

In this section, we provide detailed proofs for Theorem C.8, showing that word embeddings are insufficient for representing all videos. We begin with a 1 dimensional case, where we assume all weights in function $f(x)$ are integers.

**Lemma C.1** (Integer function bound in 1 dimension). *If the following conditions hold:*

- *Let $V \in \mathbb{N}$ denote a positive integer.*

- *Let $f : [V]^n \to \mathbb{R}$ denote a linear function where weights are all integers.*

- *Let $x \in [V]^n$ denote the input of function $f$.*

- *Let $M := \max_x f(x), m := \min_x f(x)$.*

- *Let $\epsilon = 0.5$.*

*Then we can show there exits a scalar $y \in [m, M]$ such that for any $x \in [V]^n$, $|f(x) - y| \geq \epsilon$.*

*Proof.* Since $x \in [V]^n$, all entries of $x$ are integers. Since function $f$ is a linear function where all weights are integers, the output $f(x) \in \mathcal{Z}$ can only be integer.

Therefore, $m, M \in \mathcal{Z}$. We choose $y = m + 0.5$. Since for all $f(x)$ are integers, then we have $|f(x) - y| \geq 0.5$.                                  $\square$

Then, we extend the above Lemma to $d$ dimensional case.

**Lemma C.2** (Integer function bound in $d$ dimension). *If the following conditions hold:*

- *Let $V \in \mathbb{N}$ denote a positive integer.*

- *Let $f : [V]^n \to \mathbb{R}^d$ denote a linear function where weights are all integers.*

- *Let $x \in [V]^n$ denote the input of function $f$.*

- *Let $M := \max_x \|f(x)\|_2, m := \min_x \|f(x)\|_2$.*

- *Let $\epsilon = 0.5\sqrt{d}$.*

*Then we can show there exits a vector $y \in \mathbb{R}^d$, satisfying $m \leq \|y\|_2 \leq M$, such that for any $x \in [V]^n$, $\|f(x) - y\|_2 \geq \epsilon$.*

*Proof.* Let $x_{\min} \in [V]^n$ denote the vector which satisfies $f(x_{\min}) = m$. Since all entries in $x$ and $f$ are integers, all entries in $f(x_{\min})$ are all integers.

For $i \in [d]$, let $z_i \in \mathcal{Z}$ denote the $i$-th entry of $f(x_{\min})$.

Then, we choose the vector $y \in \mathbb{R}^d$ as

$$y = \begin{bmatrix} z_1 + 0.5 \\ z_2 + 0.5 \\ \vdots \\ z_d + 0.5 \end{bmatrix}$$

Then, since all entries of $f(x)$ are integers, we have $\|f(x) - y\|_2 \geq 0.5\sqrt{d}$.        $\square$

Then, we move on to a more complicated case, in which we do not make any assumptions about the function $f(x)$. We still begin by considering the 1 dimensional case.

**Definition C.3** (Set Complement). *If the following conditions hold:*

- *Let $A, U$ denote two sets.*

*Then, we use $U \backslash A$ to denote the complement of $A$ in $U$:*

$$U \backslash A := \{x \in U : x \notin A\}$$

**Definition C.4** (Cover). *If the following conditions hold:*

- *Let $X$ denote a set.*

- *Let $A$ denote an index set.*

- *For $\alpha \in A$, let $U_\alpha \subset X$ denote the subset of $X$, indexed by $A$.*

- *Let Let $C = \{U_\alpha : \alpha \in A\}$.*

*Then we call $C$ is a cover of $X$ if the following holds:*

$$X \subseteq \cup_{\alpha \in A} U_\alpha$$

**Lemma C.5** (Any function bound in 1 dimension). *If the following conditions hold:*

- *Let $V \in \mathbb{N}$ denote a positive integer.*

- *Let $f : [V]^n \to \mathbb{R}$ denote a function.*

- *Let $x \in [V]^n$ denote the input of function $f$.*

- *Let $M := \max_x f(x), m := \min_x f(x)$.*

- *Let $\epsilon = (M - m)/(2V^n)$.*

*Then we can show there exits a scalar $y \in [m, M]$ such that for any $x \in [V]^n$, $|f(x) - y| \geq \epsilon$.*

*Proof.* Assuming for all $y \in [m, M]$, there exists one $f(x)$, such that $|f(x) - y| < (M - m)/(2V^n)$.
The overall maximum cover of all $V^n$ points should satisfy

$$2 \cdot V^n \cdot |f(x) - y| < (M - m) \tag{1}$$

where the first step follows from there are total $V^n$ possible choices for $f(x)$, and each choice has a region with length less than $2|f(x) - y|$. This is because the $y$ can be either left side of $f(x)$, or can be on the right side of $f(x)$, for both case, we need to have $|f(x) - y| < (M - m)/(2V^n)$. So the length for each region of $f(x)$ should at least be $2|f(x) - y|$.

Eq (1) indicates the overall regions of $V^n$ points can not cover all $[m, M]$ range, i.e. cannot become a cover (Definition C.4) of $[m, M]$. This is because each points can cover at most $2|f(x) - y| < (M - m)/V^n$ length, and there are total $V^n$ points. So the maximum region length is less than $V^n \cdot (M - m)/V^n = (M - m)$. Note that the length of the range $[m, M]$ is $(M - m)$. Therefore, $V^n$ points cannot cover all $[m, M]$ range.

We use $\mathcal{S}$ to denote the union of covers of all possible $f(x)$. Since the length of $\mathcal{S}$ is less than $M - m$, there exists at least one $y$ lies in $[m, M] \backslash \mathcal{S}$ such that $|f(x) - y| \geq (M - m)/(2V^n)$. Here $\backslash$ denotes the set complement operation as defined in Definition C.3.

Then, we complete our proof. $\square$

Here, we introduce an essential fact that states the volume of a $\ell_2$-ball in $d$ dimensional space.

Then, we extend our 1 dimensional result on any function $f(x)$ to $d$ dimensional cases.

**Theorem C.6** (Word embeddings are insufficient to represent for all videos, restatement of Theorem 4.9). *If the following conditions hold:*

- *Let $n, d$ denote two integers, where $n$ denotes the maximum length of the sentence, and all videos are in $\mathbb{R}^d$ space.*

- *Let $V \in \mathbb{N}$ denote the vocabulary size.*

- *Let $\mathcal{U} = \{u_1, u_2, \cdots, u_V\}$ denote the word embedding space, where for $i \in [V]$, the word embedding $u_i \in \mathbb{R}^k$.*

- *Let $\delta_{\min} = \min_{i,j \in [V], i \neq j} \|u_i - u_j\|_2$ denote the minimum $\ell_2$ distance of two word embedding.*

- *Let $f : \mathbb{R}^{nk} \to \mathbb{R}^d$ denote the mapping from sentence space (discrete space $\{u_1, \ldots, u_V\}^n$) to video space $\mathbb{R}^d$.*

- *Let $M := \max_x \|f(x)\|_2, m := \min_x \|f(x)\|_2$.*

- *Let $\epsilon = ((M^d - m^d)/V^n)^{1/d}$.*

*Then, we can show that there exits a video $y \in \mathbb{R}^d$, satisfying $m \leq \|y\|_2 \leq M$, such that for any sentence $x \in \{u_1, u_2, \cdots, u_V\}^n$, $\|f(x) - y\|_2 \geq \epsilon$.*

*Proof.* Assuming for all $y$ satisfying $m \leq \|y\|_2 \leq M$, there exists one $f(x)$, such that $|f(x) - y| < ((M^d - m^d)/V^n)^{1/d}$.

Then, according to Fact 4.3, for each $f(x)$, the volume of its cover is $\frac{\pi^{d/2}}{(d/2)!}((M^d - m^d)/V^n)$.

There are maximum total $V^n$ $f(x)$, so the maximum volume of all covers is

$$V^n \cdot \frac{\pi^{d/2}}{(d/2)!}((M^d - m^d)/V^n) < \frac{\pi^{d/2}}{(d/2)!}(M^d - m^d) \tag{2}$$

The entire space of a $d$-dimensional $\ell_2$ ball is $\frac{\pi^{d/2}}{(d/2)!}(M^d - m^d)$. However, according to Eq. (2) the maximum volume of the regions generated by all $f(x)$ is less than $\frac{\pi^{d/2}}{(d/2)!}(M^d - m^d)$. Therefore Eq. (2) indicates the cover of all $V^n$ possible points does not cover the entire space for $y$.

Therefore, there exists a $y$ satisfying $m \leq \|y\|_2 \leq M$, such that $\|f(x) - y\|_2 \geq ((M^d - m^d)/V^n)^{1/d}$.

Then, we complete our proof.

$\square$

**Definition C.7** (Bi-Lipschitzness). *We say a function $f : \mathbb{R}^n \to \mathbb{R}^d$ is $L$-bi-Lipschitz if for all $x, y \in \mathbb{R}^n$, we have*

$$L^{-1}\|x - y\|_2 \leq \|f(x) - f(y)\|_2 \leq L\|x - y\|_2.$$

Then, we state our main result as follows

**Theorem C.8** (Word embeddings are insufficient to represent for all videos, with Bi-Lipschitz condition). *If the following conditions hold:*

- *Let $n, d$ denote two integers, where $n$ denotes the maximum length of the sentence, and all videos are in $\mathbb{R}^d$ space.*

- *Let $V \in \mathbb{N}$ denote the vocabulary size.*

- *Let $\mathcal{U} = \{u_1, u_2, \cdots, u_V\}$ denote the word embedding space, where for $i \in [V]$, the word embedding $u_i \in \mathbb{R}^k$.*

- *Let $\delta_{\min} = \min_{i,j \in [V], i \neq j} \|u_i - u_j\|_2$ denote the minimum $\ell_2$ distance of two word embedding.*

- *Let $f : \mathbb{R}^{nk} \to \mathbb{R}^d$ denote the text-to-video generation model, which is also a mapping from sentence space (discrete space $\{u_1, \ldots, u_V\}^n$) to video space $\mathbb{R}^d$.*

- *Assuming $f : \mathbb{R}^{nk} \to \mathbb{R}^d$ satisfies the $L$-bi-Lipschitz condition (Definition C.7).*

- *Let $M := \max_x \|f(x)\|_2, m := \min_x \|f(x)\|_2$.*

- *Let $\epsilon = \max\{0.5 \cdot \delta_{\min}/L, ((M^d - m^d)/V^n)^{1/d}\}$.*

*Then, we can show that there exits a video $y \in \mathbb{R}^d$, satisfying $m \leq \|y\|_2 \leq M$, such that for any sentence $x \in \{u_1, u_2, \cdots, u_V\}^n$, $\|f(x) - y\|_2 \geq \epsilon$.*

*Proof.* Our goal is to prove that when the bi-Lipschitz condition (Definition C.7) holds for $f(x)$, the statement can be held with $\epsilon = \max\{0.5 \cdot \delta_{\min}/L, ((M^d - m^d)/V^n)^{1/d}\}$.

According to Lemma 4.9, we have that $\epsilon \geq ((M^d - m^d)/V^n)^{1/d}$. Then, we only need to prove that when $0.5 \cdot \delta_{\min}/L > ((M^d - m^d)/V^n)^{1/d}$, holds, $\epsilon = \max\{0.5 \cdot \delta_{\min}/L, ((M^d - m^d)/V^n)^{1/d}\} = 0.5 \cdot \delta_{\min}$, our statement still holds.

Since we have assume that the function $f(x)$ satisfies that for all $x, y \in \mathbb{R}^{nk}$, such that

$$\|f(x) - f(y)\|_2 \geq \|x - y\|_2/L. \tag{3}$$

According to the definition of $\delta_{\min}$, we have for all $i, j \in [V], i \neq j$, such that

$$\|u_i - u_j\|_2 \geq \delta_{\min} \tag{4}$$

Combining Eq. (3) and (4), we have for all $i, j \in [V], i \neq j$

$$\|f(u_i) - f(u_j)\|_2 \geq \delta_{\min}/L \tag{5}$$

We choose $y = f(\frac{1}{2}(u_i + u_j))$ for any $i, j \in [V], i \neq j$

Then, for all $k \in [V]$, we have

$$\|y - f(u_i)\|_2 \geq \|\frac{1}{2}(u_i + u_j) - u_k\|_2/L$$
$$\geq 0.5 \cdot \delta_{\min}/L$$

where the first step follows from $f(x)$ satisfies the bi-Lipschitz condition, the second step follows from Eq. (5).

Therefore, when we have $0.5 \cdot \delta_{\min}/L > ((M^d - m^d)/V^n)^{1/d}$ holds, then we must have $\epsilon = 0.5 \cdot \delta_{\min}/L$.

Considering all conditions we discussed above, we are safe to conclude that $\epsilon = \max\{0.5 \cdot \delta_{\min}/L, ((M^d - m^d)/V^n)^{1/d}\}$ □

Table 2: **Statement Reference Table.** This table shows the relationship between definitions and algorithms used in the paper, helping readers easily track where each term is defined and referenced.

| Statements | Comment | Call | Called by |
|---|---|---|---|
| Def. 4.1 | Define linear interpolation | None | Alg. 2, Alg. 1 |
| Def. 4.2 | Define cosine similarity calculator | None | Alg. 2, Alg. 1 |
| Def. 4.4 | Define attention layer | None | Alg. 4, Def. 4.7 |
| Def. 4.5 | Define convolution layer | None | Alg. 4, Def. 4.7 |
| Def. 4.6 | Define linear projection | None | Alg. 4, Def. 4.7 |
| Def. 4.7 | Define 3D attention | Def. 4.4, Def. 4.5, Def. 4.6 | Alg. 4, Def. 4.8 |
| Def. 4.8 | Define text to video generation model | Def. 4.7 | Def. 2.1 |
| Def. 2.1 | Define optimal interpolation embedding | Def. 4.8 | Alg. 3 |
| Alg. 4 | 3D Attention algorithm | Def. 4.4, Def. 4.5, Def. 4.6, Def. 4.7 | None |
| Alg. 2 | Cosine similarity calculator algorithm | Def. 4.1, Def. 4.2 | Alg. 1 |
| Alg. 1 | Find optimal interpolation algorithm | Def. 4.1, Def. 4.2, Alg. 2 | Alg. 3 |
| Alg. 3 | Video interpolation algorithm | Alg. 1 | None |

Prompt A: *"The **tiger**, moves gracefully through the forest, its fur flowing in the breeze."*

Step:

$i = 1$

$i = 16$

$i = 17$

$i = 18$

$i = 30$

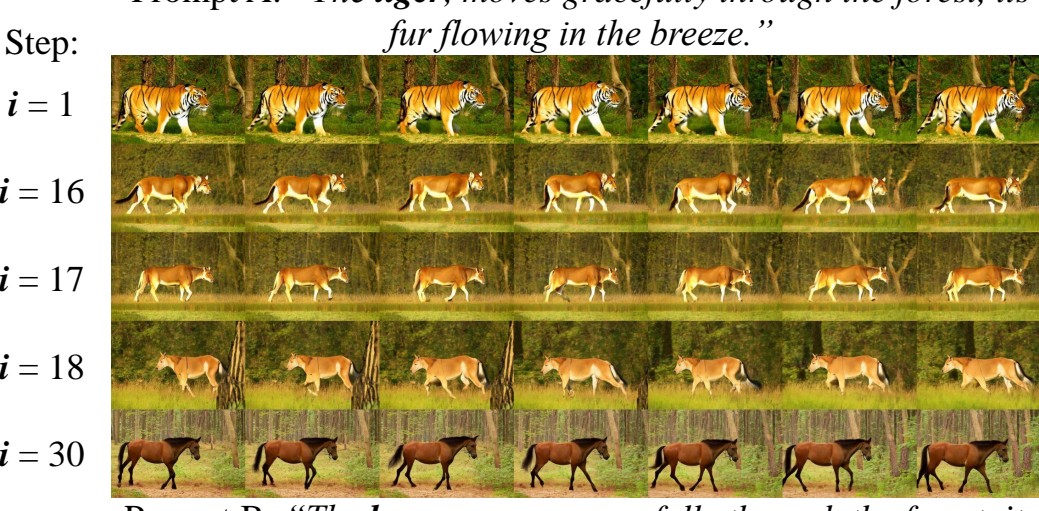

Prompt B: *"The **horse**, moves gracefully through the forest, its fur flowing in the breeze."*

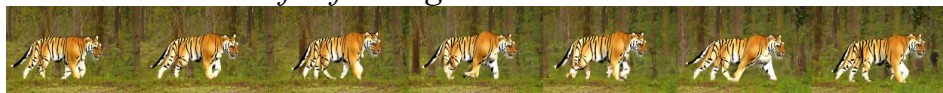

Prompt C: *"The tiger, **which has horse legs and no black strips on its fur**, moves gracefully through the forest, its fur flowing in the breeze."*

Figure 5: **Mixture of ["Tiger"] and ["Horse"]**. Our objective is to mix the features described in Prompt A and Prompt B with the guidance of Prompt C. We set the total number of interpolation steps to 30. Using Algorithm 1, we identify the 17-th interpolation embedding as the optimal embedding and generate the corresponding video. The video generated directly from Prompt C does not exhibit the desired mixed features from Prompts A and B.

## D    MORE EXAMPLES

In this section, we will show more experimental results that the video generated directly from the guidance prompt does not exhibit the desired mixed features from the prompts.

## E    FULL ALGORITHM

In this section, we provide the algorithm for 3D attention in Algorithm 4.

Prompt A: *"The **eggplant**, freshly washed, served in a dish, is placed on the wooden table."*

Step:

$i = 1$

$i = 8$

$i = 9$

$i = 10$

$i = 30$

Prompt B: *"The **orange**, freshly washed, served in a dish, is placed on the wooden table."*

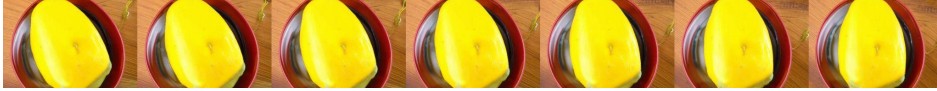

Prompt C: *"The eggplant, **with the color of yellow**, freshly washed, served in a dish, is placed on the wooden table."*

Figure 6: **Mixture of ["Eggplant"] and ["Orange"]**. Our objective is to mix the features described in Prompt A and Prompt B with the guidance of Prompt C. We set the total number of interpolation steps to 30. Using Algorithm 1, we identify the 9-th interpolation embedding as the optimal embedding and generate the corresponding video. The video generated directly from Prompt C does not exhibit the desired mixed features from Prompts A and B.

Prompt A: *"The **airplane** landed gently on the runway, its wheels touching the ground with precision."*

Step:

$i = 1$

$i = 15$

$i = 16$

$i = 17$

$i = 30$

Prompt B: *"The **horse** stopped gracefully at the water's edge, its reflection shimmering in the pond."*

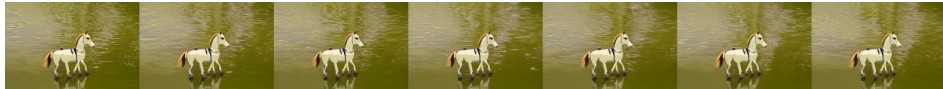

Prompt C: *"The **robot horse** stopped gracefully at the water's edge, its reflection shimmering in the pond."*

Figure 7: **Mixture of ["Airplane"] and ["Horse"].** Our objective is to mix the features described in Prompt A and Prompt B with the guidance of Prompt C. We set the total number of interpolation steps to 30. Using Algorithm 1, we identify the 16-th interpolation embedding as the optimal embedding and generate the corresponding video. The video generated directly from Prompt C does not exhibit the desired mixed features from Prompts A and B.

Prompt A: *"An **airplane** soared above the clouds, its engines humming as it crossed the horizon."*

Step:

$i = 1$

$i = 14$

$i = 15$

$i = 16$

$i = 30$

Prompt B: *"An **automobile** drove along the winding mountain road, its engine purring smoothly."*

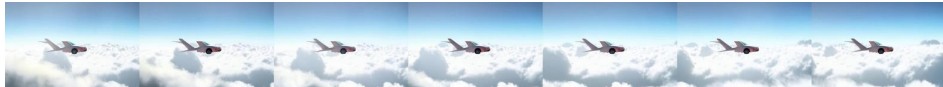

Prompt C: *"An **automobile with airplane wings** soared above the clouds, its engines humming as it crossed the horizon."*

Figure 8: **Mixture of ["Airplane"] and ["Automobile"]**. Our objective is to mix the features described in Prompt A and Prompt B with the guidance of Prompt C. We set the total number of interpolation steps to 30. Using Algorithm 1, we identify the 15-th interpolation embedding as the optimal embedding and generate the corresponding video. The video generated directly from Prompt C does not exhibit the desired mixed features from Prompts A and B.

Prompt A: *"The **airplane** took off with a roar, lifting off the ground as it climbed into the sky.."*

Step:

$i = 1$

$i = 13$

$i = 14$

$i = 15$

$i = 30$

Prompt B: *"The **dog** barked excitedly at the door, wagging its tail in anticipation of a walk."*

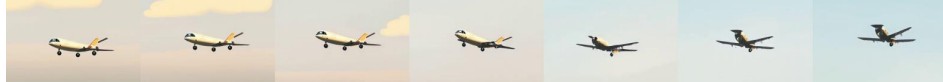

Prompt C: *"The airplane **with a dog head** took off with a roar, lifting off the ground as it climbed into the sky."*

Figure 9: **Mixture of ["Airplane"] and ["Dog"]**. Our objective is to mix the features described in Prompt A and Prompt B with the guidance of Prompt C. We set the total number of interpolation steps to 30. Using Algorithm 1, we identify the 14-th interpolation embedding as the optimal embedding and generate the corresponding video. The video generated directly from Prompt C does not exhibit the desired mixed features from Prompts A and B.

Prompt A: *"An **airplane** glided smoothly in the sky."*

Step:

$i = 1$

$i = 14$

$i = 15$

$i = 16$

$i = 30$

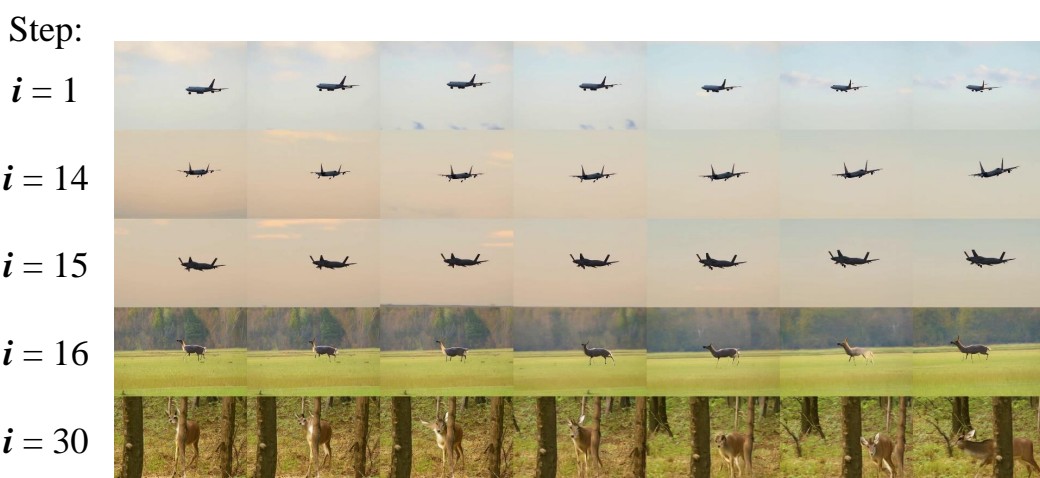

Prompt B: *"A **deer** moved quietly in the woods."*

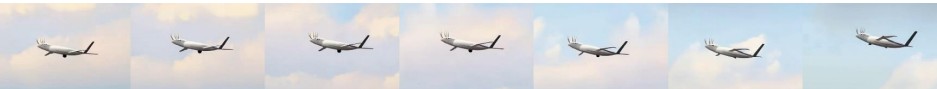

Prompt C: *"An **airplane with antlers**, glided smoothly in the sky."*

Figure 10: **Mixture of ["Airplane"] and ["Deer"]**. Our objective is to mix the features described in Prompt A and Prompt B with the guidance of Prompt C. We set the total number of interpolation steps to 30. Using Algorithm 1, we identify the 15-th interpolation embedding as the optimal embedding and generate the corresponding video. The video generated directly from Prompt C does not exhibit the desired mixed features from Prompts A and B.

Prompt A: *"The **airplane** rested on the runway."*

Step:

$i = 1$

$i = 16$

$i = 17$

$i = 18$

$i = 30$

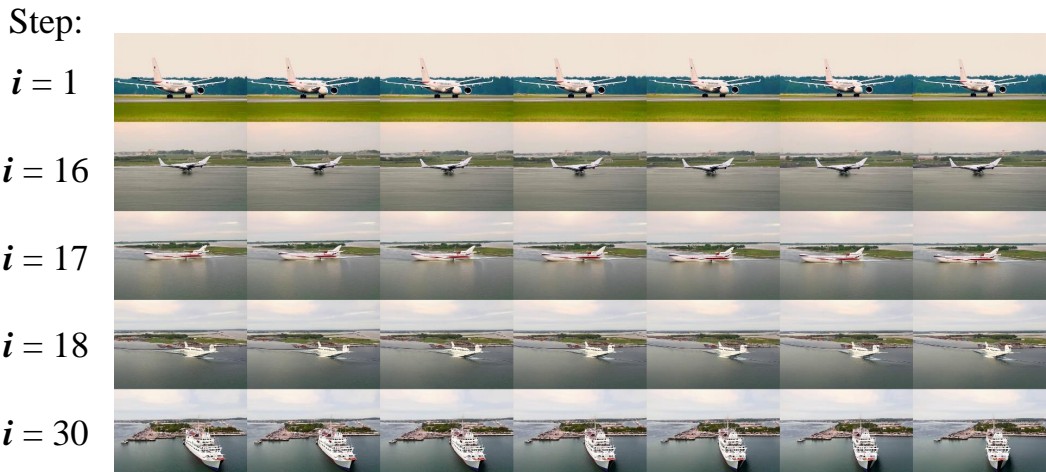

Prompt B: *"The **ship** anchored at the harbor."*

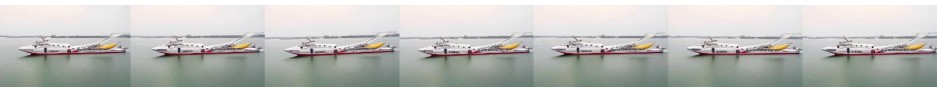

Prompt C: *"The **ship with the shape of airplane**, anchored at the harbor."*

Figure 11: **Mixture of ["Airplane"] and ["Ship"]**. Our objective is to mix the features described in Prompt A and Prompt B with the guidance of Prompt C. We set the total number of interpolation steps to 30. Using Algorithm 1, we identify the 17-th interpolation embedding as the optimal embedding and generate the corresponding video. The video generated directly from Prompt C does not exhibit the desired mixed features from Prompts A and B.

Prompt A: *"The **airplane** rested on the runway."*

Step:

$i = 1$

$i = 16$

$i = 17$

$i = 18$

$i = 30$

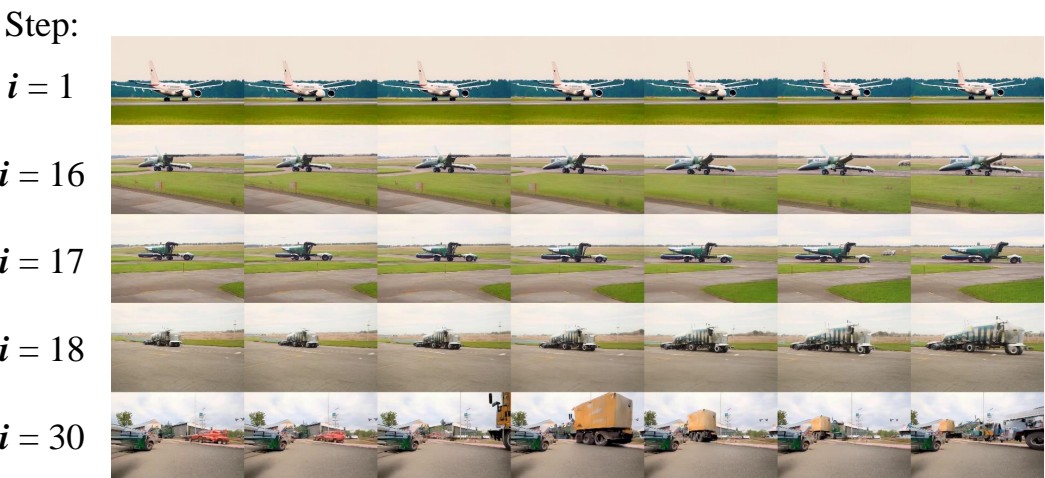

Prompt B: *"The **truck** parked at the station."*

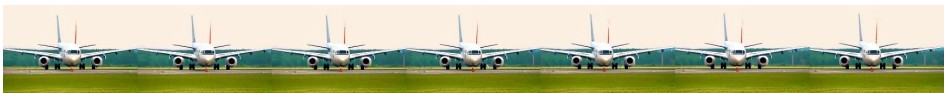

Prompt C: *"The **airplane with the shape of truck**, rested on the runway."*

Figure 12: **Mixture of ["Airplane"] and ["Truck"]**. Our objective is to mix the features described in Prompt A and Prompt B with the guidance of Prompt C. We set the total number of interpolation steps to 30. Using Algorithm 1, we identify the 17-th interpolation embedding as the optimal embedding and generate the corresponding video. The video generated directly from Prompt C does not exhibit the desired mixed features from Prompts A and B.

Prompt A: *"The **automobile** parked by the forest edge."*

Step:

$i = 1$

$i = 14$

$i = 15$

$i = 16$

$i = 30$

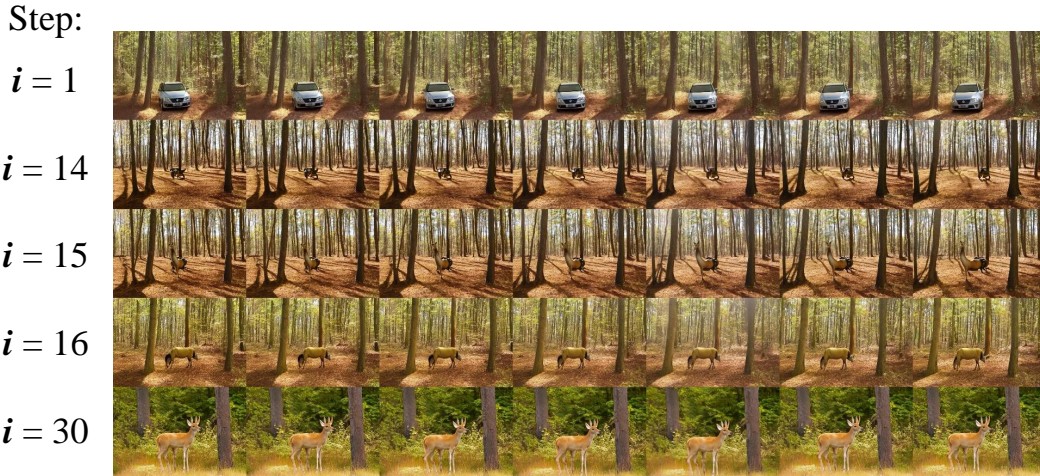

Prompt B: *"The **deer** stood quietly at the forest edge."*

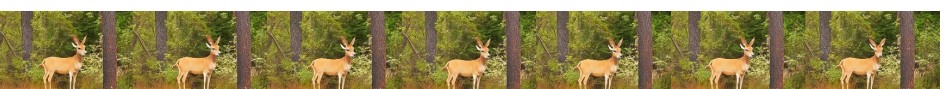

Prompt C: *"The **deer with the body of a car**, stood quietly at the forest edge."*

Figure 13: **Mixture of ["Automobile"] and ["Deer"]**. Our objective is to mix the features described in Prompt A and Prompt B with the guidance of Prompt C. We set the total number of interpolation steps to 30. Using Algorithm 1, we identify the 15-th interpolation embedding as the optimal embedding and generate the corresponding video. The video generated directly from Prompt C does not exhibit the desired mixed features from Prompts A and B.

Prompt A: *"The **automobile** parked by the barn."*

Step:

$i = 1$

$i = 15$

$i = 16$

$i = 17$

$i = 30$

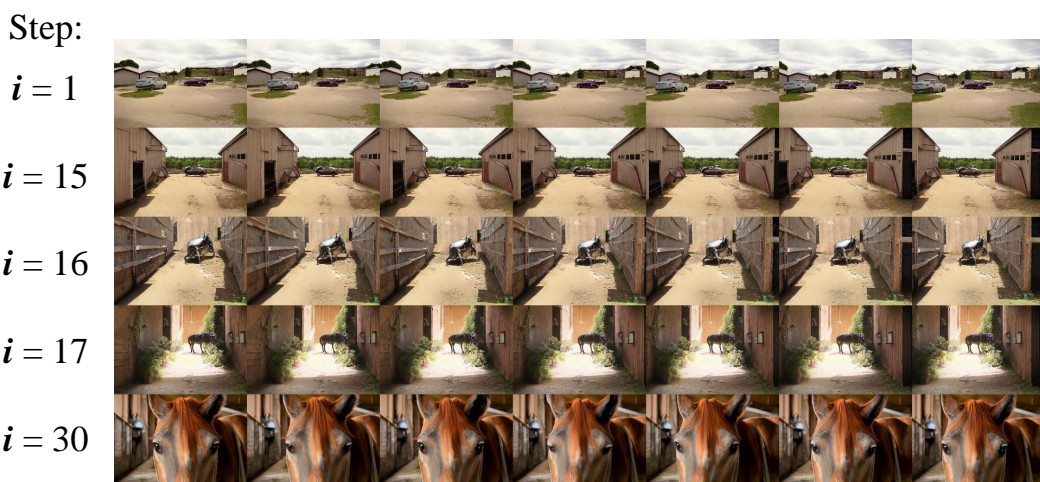

Prompt B: *"The **horse** stood quietly in the stable."*

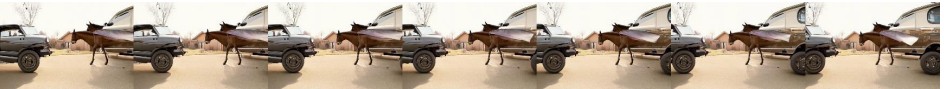

Prompt C: *"The **automobile with four horse legs**, parked by the barn."*

Figure 14: **Mixture of ["Automobile"] and ["Horse"]**. Our objective is to mix the features described in Prompt A and Prompt B with the guidance of Prompt C. We set the total number of interpolation steps to 30. Using Algorithm 1, we identify the 16-th interpolation embedding as the optimal embedding and generate the corresponding video. The video generated directly from Prompt C does not exhibit the desired mixed features from Prompts A and B.

Prompt A: *"The **automobile** parked by the beach."*

Step:

$i = 1$

$i = 14$

$i = 15$

$i = 16$

$i = 30$

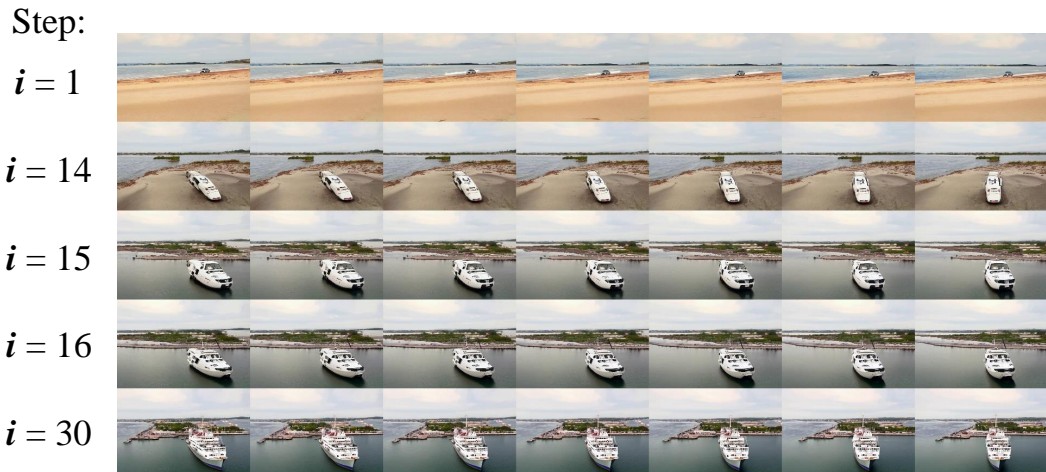

Prompt B: *"The **ship** anchored at the harbor."*

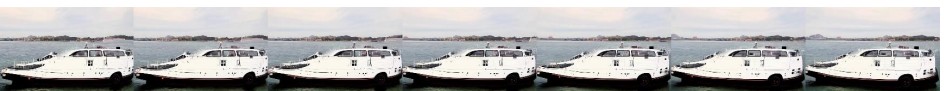

Prompt C: *"The **ship with the shape of the automobile**,*
*anchored at the harbor."*

Figure 15: **Mixture of ["Automobile"] and ["Ship"]**. Our objective is to mix the features described in Prompt A and Prompt B with the guidance of Prompt C. We set the total number of interpolation steps to 30. Using Algorithm 1, we identify the 15-th interpolation embedding as the optimal embedding and generate the corresponding video. The video generated directly from Prompt C does not exhibit the desired mixed features from Prompts A and B.

Prompt A: *"A **bird** perched on a tree branch."*

Step:

$i = 1$

$i = 16$

$i = 17$

$i = 18$

$i = 30$

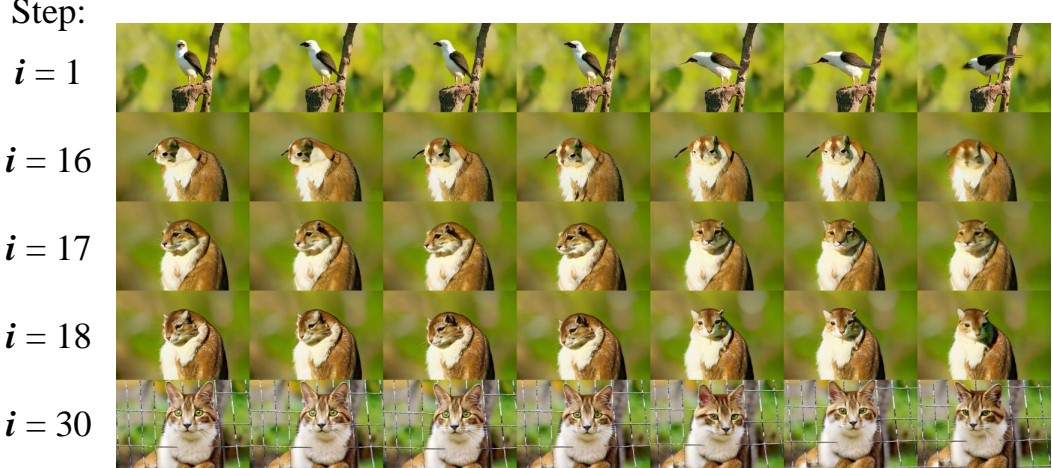

Prompt B: *"A **cat** sat on the fence nearby."*

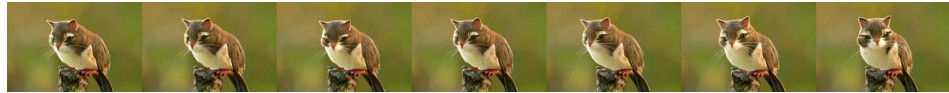

Prompt C: *"A **bird with a cat head**, perched on a tree branch."*

Figure 16: **Mixture of ["Bird"] and ["Cat"]**. Our objective is to mix the features described in Prompt A and Prompt B with the guidance of Prompt C. We set the total number of interpolation steps to 30. Using Algorithm 1, we identify the 17-th interpolation embedding as the optimal embedding and generate the corresponding video. The video generated directly from Prompt C does not exhibit the desired mixed features from Prompts A and B.

Prompt A: *"The **bird** perched on a branch."*

Step:

$i = 1$

$i = 11$

$i = 12$

$i = 13$

$i = 30$

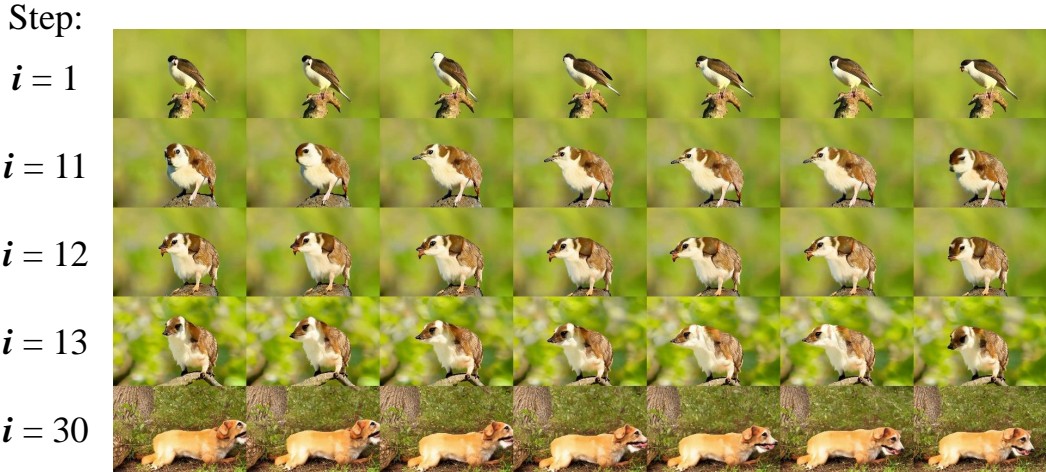

Prompt B: *"The **dog** lay under the tree."*

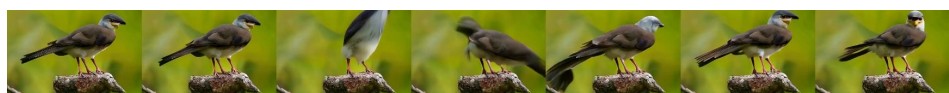

Prompt C: *"A **bird with four legs**, perched on a tree branch."*

Figure 17: **Mixture of ["Bird"] and ["Dog"]**. Our objective is to mix the features described in Prompt A and Prompt B with the guidance of Prompt C. We set the total number of interpolation steps to 30. Using Algorithm 1, we identify the 12-th interpolation embedding as the optimal embedding and generate the corresponding video. The video generated directly from Prompt C does not exhibit the desired mixed features from Prompts A and B.

Prompt A: *"The **bird** perched on a branch."*

Step:

$i = 1$

$i = 15$

$i = 16$

$i = 17$

$i = 30$

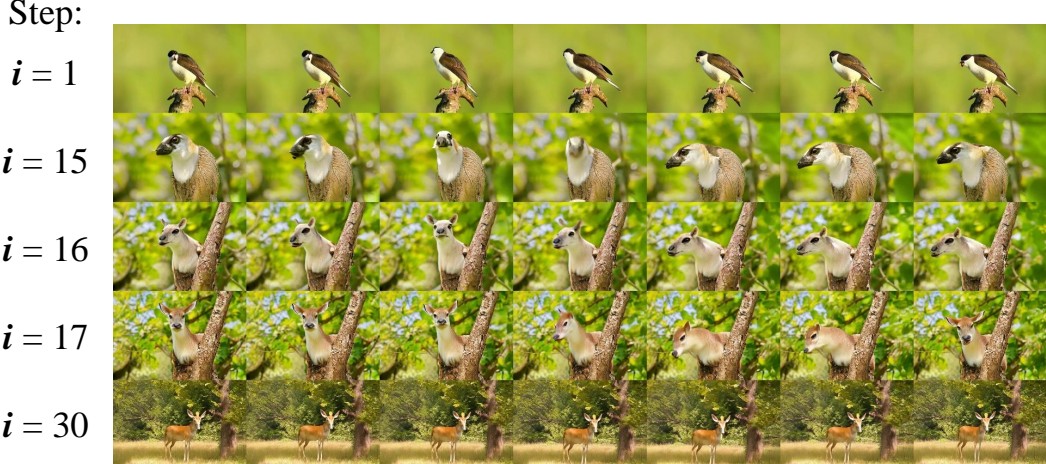

Prompt B: *"The **deer** stood under the tree."*

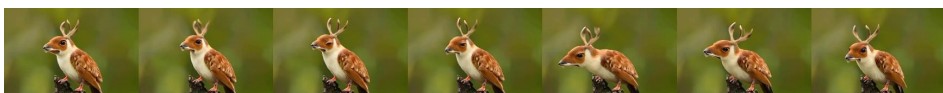

Prompt C: *"A **bird with a deer head**, perched on a tree branch."*

Figure 18: **Mixture of ["Bird"] and ["Deer"]**. Our objective is to mix the features described in Prompt A and Prompt B with the guidance of Prompt C. We set the total number of interpolation steps to 30. Using Algorithm 1, we identify the 16-th interpolation embedding as the optimal embedding and generate the corresponding video. The video generated directly from Prompt C does not exhibit the desired mixed features from Prompts A and B.

Prompt A: *"An **elephant** walked gracefully through the savanna."*

Step:

$i = 1$

$i = 15$

$i = 16$

$i = 17$

$i = 30$

Prompt B: *"A **lion** prowled silently through the savanna."*

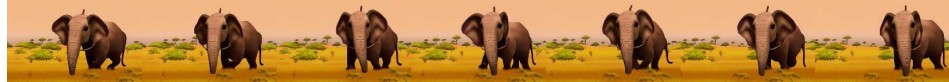

Prompt C: *"An **elephant with a face of lion**, walked gracefully through the savanna."*

Figure 19: **Mixture of ["Elephant"] and ["Lion"]**. Our objective is to mix the features described in Prompt A and Prompt B with the guidance of Prompt C. We set the total number of interpolation steps to 30. Using Algorithm 1, we identify the 16-th interpolation embedding as the optimal embedding and generate the corresponding video. The video generated directly from Prompt C does not exhibit the desired mixed features from Prompts A and B.

Prompt A: *"An **orchid** bloomed gracefully in the greenhouse."*

Step:

$i = 1$

$i = 14$

$i = 15$

$i = 16$

$i = 30$

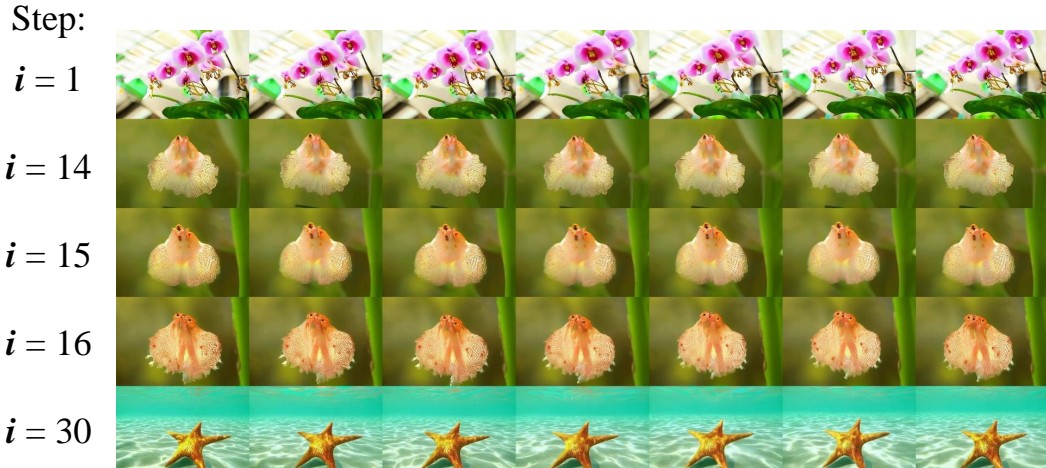

Prompt B: *"A **starfish** rested quietly on the ocean floor."*

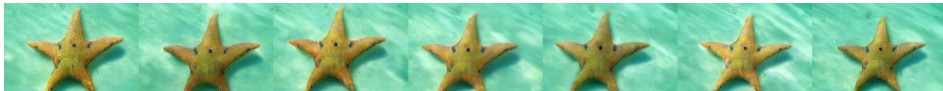

Prompt C: *"A **starfish with the shape of orchid**, rested quietly on the ocean floor."*

Figure 20: **Mixture of ["Orchid"] and ["Starfish"].** Our objective is to mix the features described in Prompt A and Prompt B with the guidance of Prompt C. We set the total number of interpolation steps to 30. Using Algorithm 1, we identify the 15-th interpolation embedding as the optimal embedding and generate the corresponding video. The video generated directly from Prompt C does not exhibit the desired mixed features from Prompts A and B.

Prompt A: *"A **sunflower** bloomed brightly in the summer field."*

Step:

$i = 1$

$i = 19$

$i = 20$

$i = 21$

$i = 30$

Prompt B: *"A **starfish** rested quietly on the ocean floor."*

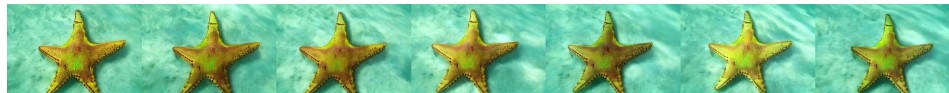

Prompt C: *"A **starfish with the shape and color of sunflower**, rested quietly on the ocean floor."*

Figure 21: **Mixture of ["Sunflower"] and ["Starfish"].** Our objective is to mix the features described in Prompt A and Prompt B with the guidance of Prompt C. We set the total number of interpolation steps to 30. Using Algorithm 1, we identify the 20-th interpolation embedding as the optimal embedding and generate the corresponding video. The video generated directly from Prompt C does not exhibit the desired mixed features from Prompts A and B.

Prompt A: *"A **sunflower** stood tall in the garden."*

Step:

$i = 1$

$i = 18$

$i = 19$

$i = 20$

$i = 30$

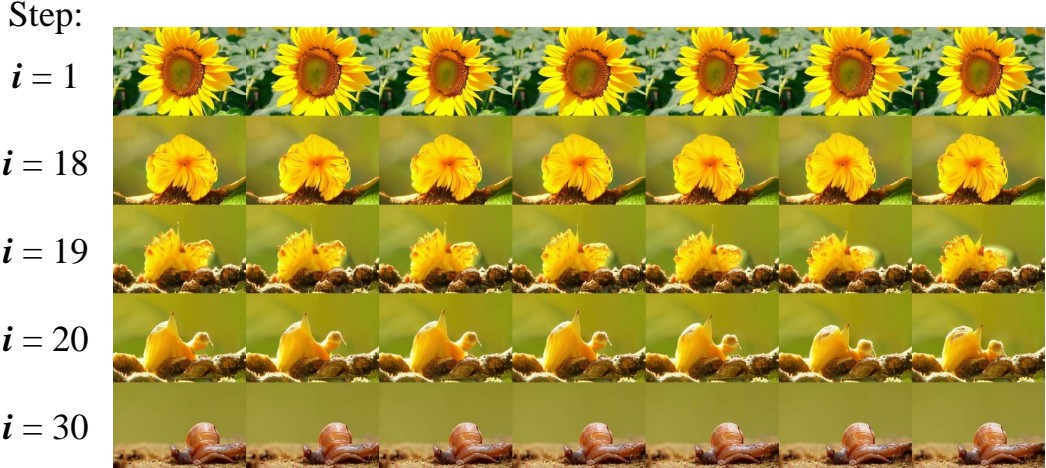

Prompt B: *"A **snail** crawled slowly across the ground."*

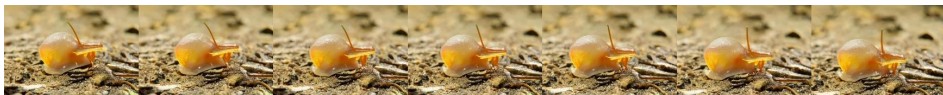

Prompt C: *"A **snail with the shape and color of sunflower**, crawled slowly across the ground."*

Figure 22: **Mixture of ["Sunflower"] and ["Snail"]**. Our objective is to mix the features described in Prompt A and Prompt B with the guidance of Prompt C. We set the total number of interpolation steps to 30. Using Algorithm 1, we identify the 19-th interpolation embedding as the optimal embedding and generate the corresponding video. The video generated directly from Prompt C does not exhibit the desired mixed features from Prompts A and B.

Prompt A: *"A **sunflower** stood tall in the bright field."*

Step:

$i = 1$

$i = 16$

$i = 17$

$i = 18$

$i = 30$

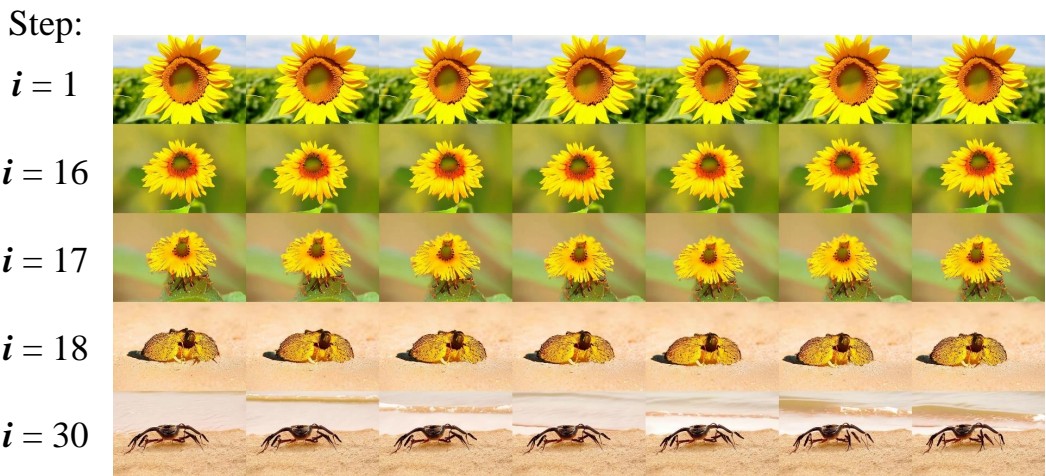

Prompt B: *"A **crab** laid on the sandy beach."*

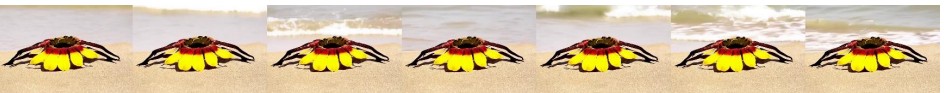

Prompt C: *"A **crab with the shape of sunflower**,*
*laid on the sandy beach."*

Figure 23: **Mixture of ["Sunflower"] and ["Crab"]**. Our objective is to mix the features described in Prompt A and Prompt B with the guidance of Prompt C. We set the total number of interpolation steps to 30. Using Algorithm 1, we identify the 17-th interpolation embedding as the optimal embedding and generate the corresponding video. The video generated directly from Prompt C does not exhibit the desired mixed features from Prompts A and B.

Prompt A: *"A **bird** perched on a tree branch."*

Step:

$i = 1$

$i = 15$

$i = 16$

$i = 17$

$i = 30$

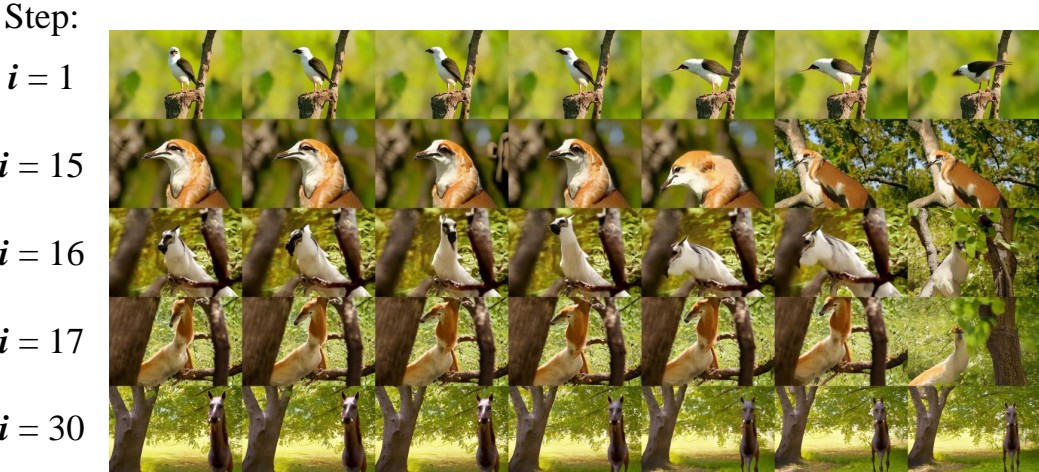

Prompt B: *"A **horse** stood under the tree."*

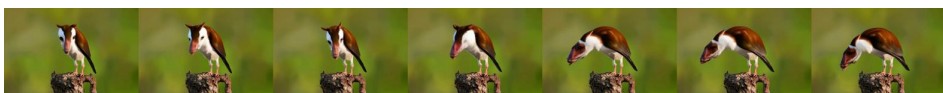

Prompt C: *"A **bird with a horse head**, perched on a tree branch."*

Figure 24: **Mixture of ["Bird"] and ["Horse"]**. Our objective is to mix the features described in Prompt A and Prompt B with the guidance of Prompt C. We set the total number of interpolation steps to 30. Using Algorithm 1, we identify the 16-th interpolation embedding as the optimal embedding and generate the corresponding video. The video generated directly from Prompt C does not exhibit the desired mixed features from Prompts A and B.

Prompt A: *"A **butterfly** landed delicately on a vibrant petal."*

Step:

$i = 1$

$i = 15$

$i = 16$

$i = 17$

$i = 30$

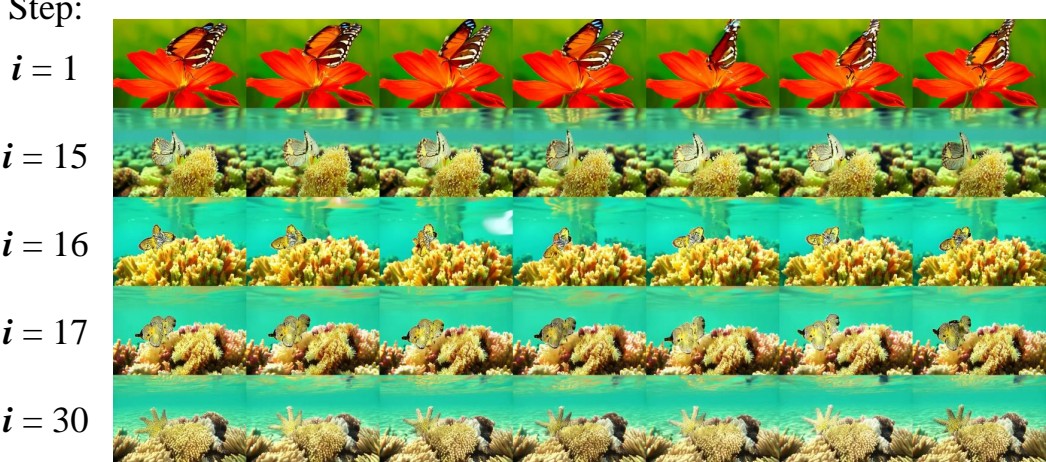

Prompt B: *"A **starfish** clung to a coral reef beneath the waves."*

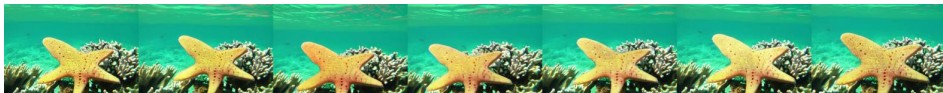

Prompt C: *"A **starfish with the shape of butterfly**, clung to a coral reef beneath the waves."*

Figure 25: **Mixture of ["Butterfly"] and ["Starfish"]**. Our objective is to mix the features described in Prompt A and Prompt B with the guidance of Prompt C. We set the total number of interpolation steps to 30. Using Algorithm 1, we identify the 16-th interpolation embedding as the optimal embedding and generate the corresponding video. The video generated directly from Prompt C does not exhibit the desired mixed features from Prompts A and B.

Prompt A: *"A **cat** stretched lazily under the sun."*

Step:

$i = 1$

$i = 14$

$i = 15$

$i = 16$

$i = 30$

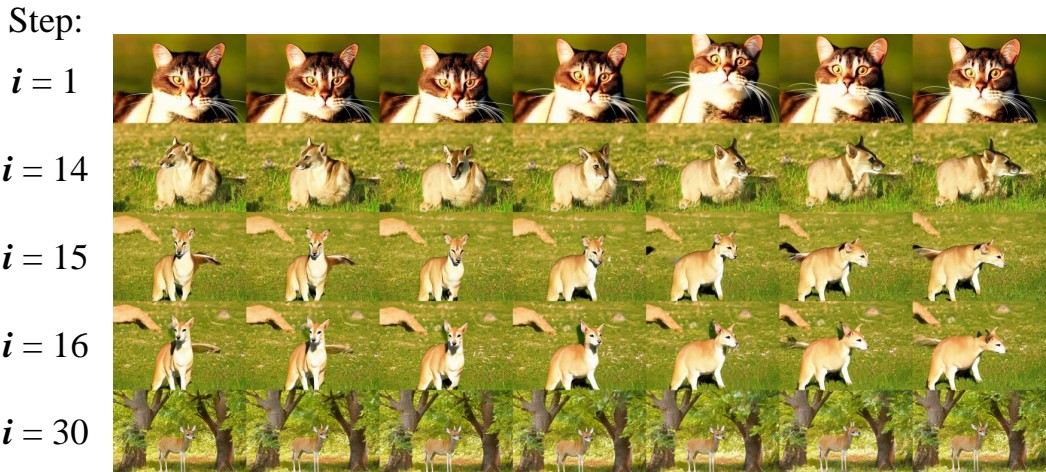

Prompt B: *"A **deer** rested peacefully under the tree."*

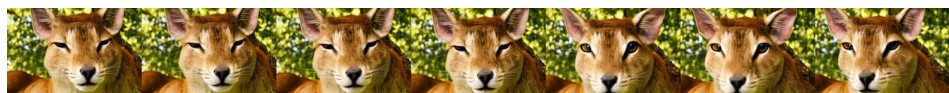

Prompt C: *"A **deer with a cat face**, rested peacefully under the tree."*

Figure 26: **Mixture of ["Cat"] and ["Deer"]**. Our objective is to mix the features described in Prompt A and Prompt B with the guidance of Prompt C. We set the total number of interpolation steps to 30. Using Algorithm 1, we identify the 15-th interpolation embedding as the optimal embedding and generate the corresponding video. The video generated directly from Prompt C does not exhibit the desired mixed features from Prompts A and B.

Prompt A: *"A **cat** sat quietly on the windowsill."*

Step:

$i = 1$

$i = 15$

$i = 16$

$i = 17$

$i = 30$

Prompt B: *"A **dog** lay quietly on the porch."*

Prompt C: *"A **cat with a dog face**, sat quietly on the windowsill."*

Figure 27: **Mixture of ["Cat"] and ["Dog"]**. Our objective is to mix the features described in Prompt A and Prompt B with the guidance of Prompt C. We set the total number of interpolation steps to 30. Using Algorithm 1, we identify the 16-th interpolation embedding as the optimal embedding and generate the corresponding video. The video generated directly from Prompt C does not exhibit the desired mixed features from Prompts A and B.

Prompt A: *"The **cat** stretched lazily under the sun."*

Step:

$i = 1$

$i = 12$

$i = 13$

$i = 14$

$i = 30$

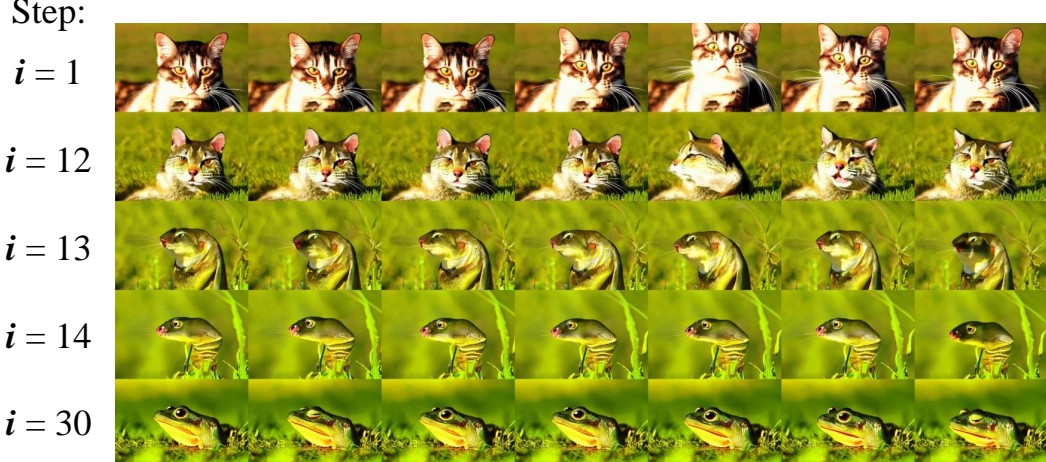

Prompt B: *"The **frog** basked quietly under the sun."*

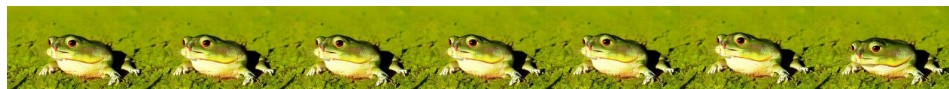

Prompt C: *"The **frog with a cat face**, basked quietly under the sun."*

Figure 28: **Mixture of ["Cat"] and ["Frog"]**. Our objective is to mix the features described in Prompt A and Prompt B with the guidance of Prompt C. We set the total number of interpolation steps to 30. Using Algorithm 1, we identify the 13-th interpolation embedding as the optimal embedding and generate the corresponding video. The video generated directly from Prompt C does not exhibit the desired mixed features from Prompts A and B.

Prompt A: *"The **cat** stretched lazily in the sun."*

Step:

$i = 1$

$i = 10$

$i = 11$

$i = 12$

$i = 30$

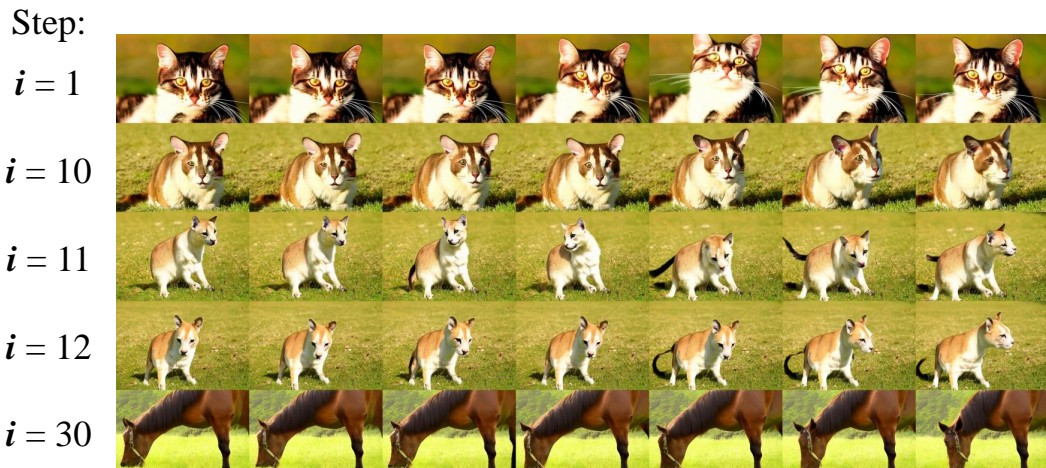

Prompt B: *"The **horse** grazed peacefully in the meadow."*

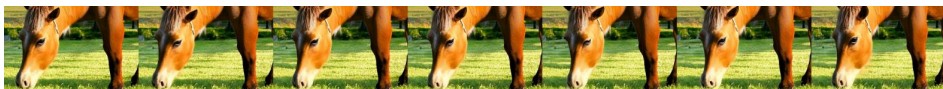

Prompt C: *"The **horse with a cat face**, grazed peacefully in the meadow."*

Figure 29: **Mixture of ["Cat"] and ["Horse"]**. Our objective is to mix the features described in Prompt A and Prompt B with the guidance of Prompt C. We set the total number of interpolation steps to 30. Using Algorithm 1, we identify the 11-th interpolation embedding as the optimal embedding and generate the corresponding video. The video generated directly from Prompt C does not exhibit the desired mixed features from Prompts A and B.

Prompt A: *"The **otter** rested on a smooth stone by the water."*

Step:

$i = 1$

$i = 13$

$i = 14$

$i = 15$

$i = 30$

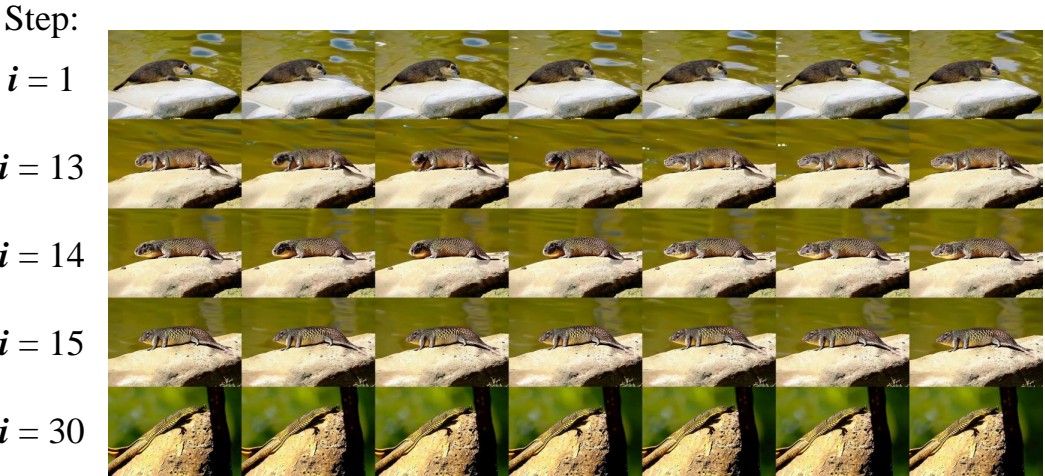

Prompt B: *"The **lizard** basked on the tree bark under the sun."*

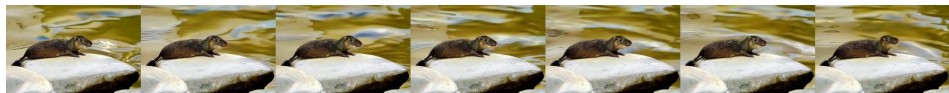

Prompt C: *"The **otter with lizard skin**, rested on a smooth stone by the water."*

Figure 30: **Mixture of ["Otter"] and ["Lizard"]**. Our objective is to mix the features described in Prompt A and Prompt B with the guidance of Prompt C. We set the total number of interpolation steps to 30. Using Algorithm 1, we identify the 14-th interpolation embedding as the optimal embedding and generate the corresponding video. The video generated directly from Prompt C does not exhibit the desired mixed features from Prompts A and B.

Prompt A: *"A **kangaroo** rested in the shade of a tall tree."*

Step:

$i = 1$

$i = 15$

$i = 16$

$i = 17$

$i = 30$

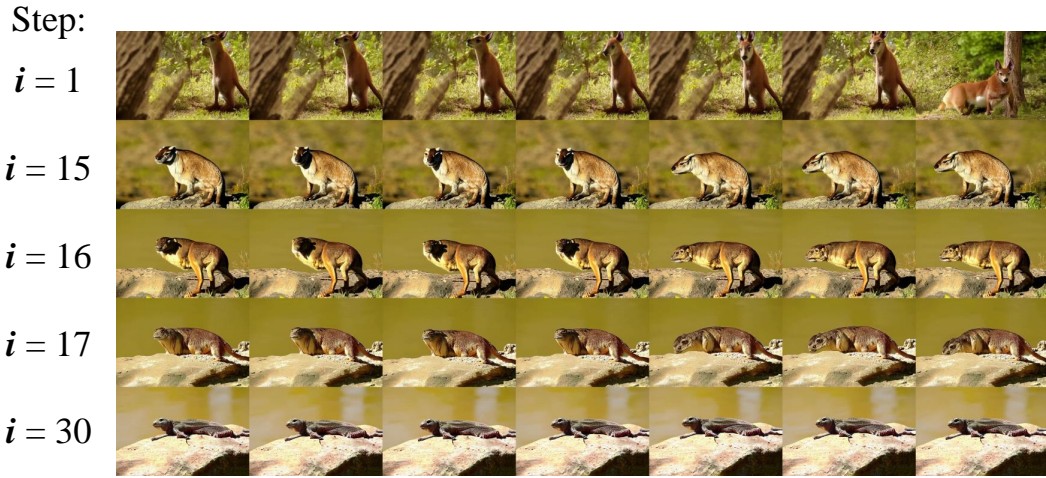

Prompt B: *"A **lizard** basked in the sunlight on a flat rock."*

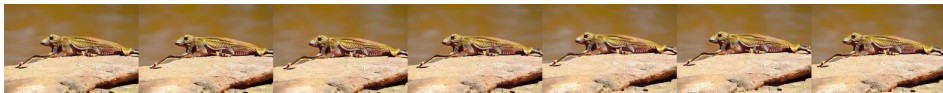

Prompt C: *"A **lizard with kangaroo legs**, basked in the sunlight on a flat rock."*

Figure 31: **Mixture of ["Kangaroo"] and ["Lizard"]**. Our objective is to mix the features described in Prompt A and Prompt B with the guidance of Prompt C. We set the total number of interpolation steps to 30. Using Algorithm 1, we identify the 16-th interpolation embedding as the optimal embedding and generate the corresponding video. The video generated directly from Prompt C does not exhibit the desired mixed features from Prompts A and B.

Prompt A: *"A **deer** stood quietly in the meadow."*

Step:

$i = 1$

$i = 16$

$i = 17$

$i = 18$

$i = 30$

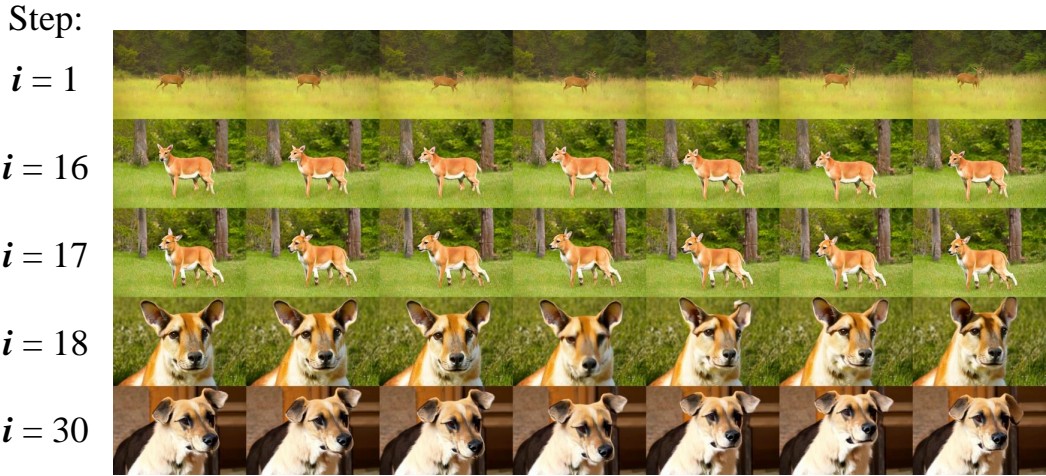

Prompt B: *"A **dog** sat quietly on the porch."*

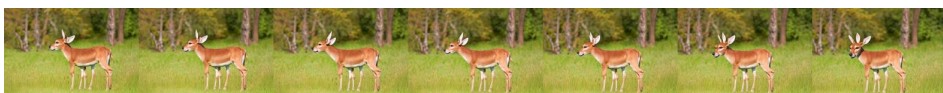

Prompt C: *"A **deer with a dog face**, stood quietly in the meadow."*

Figure 32: **Mixture of ["Deer"] and ["Dog"]**. Our objective is to mix the features described in Prompt A and Prompt B with the guidance of Prompt C. We set the total number of interpolation steps to 30. Using Algorithm 1, we identify the 17-th interpolation embedding as the optimal embedding and generate the corresponding video. The video generated directly from Prompt C does not exhibit the desired mixed features from Prompts A and B.

Prompt A: *"A **dog** lay lazily in the sun."*

Step:

$i = 1$

$i = 15$

$i = 16$

$i = 17$

$i = 30$

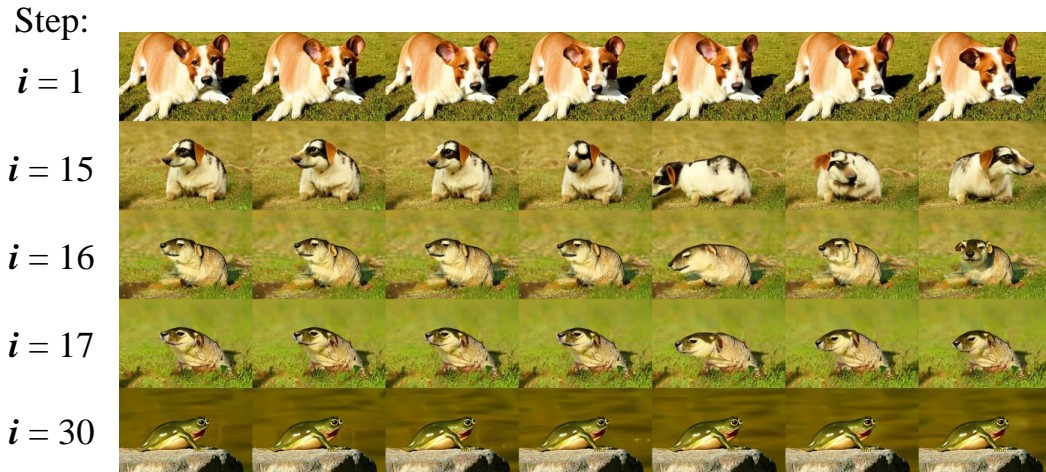

Prompt B: *"A **frog** sat still on a rock in the sun."*

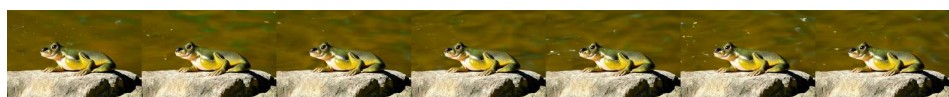

Prompt C: *"A **frog with a dog head**, sat still on a rock in the sun."*

Figure 33: **Mixture of ["Dog"] and ["Frog"]**. Our objective is to mix the features described in Prompt A and Prompt B with the guidance of Prompt C. We set the total number of interpolation steps to 30. Using Algorithm 1, we identify the 16-th interpolation embedding as the optimal embedding and generate the corresponding video. The video generated directly from Prompt C does not exhibit the desired mixed features from Prompts A and B.

Prompt A: *"The **red panda** rested on a sturdy branch."*

Step:

$i = 1$

$i = 14$

$i = 15$

$i = 16$

$i = 30$

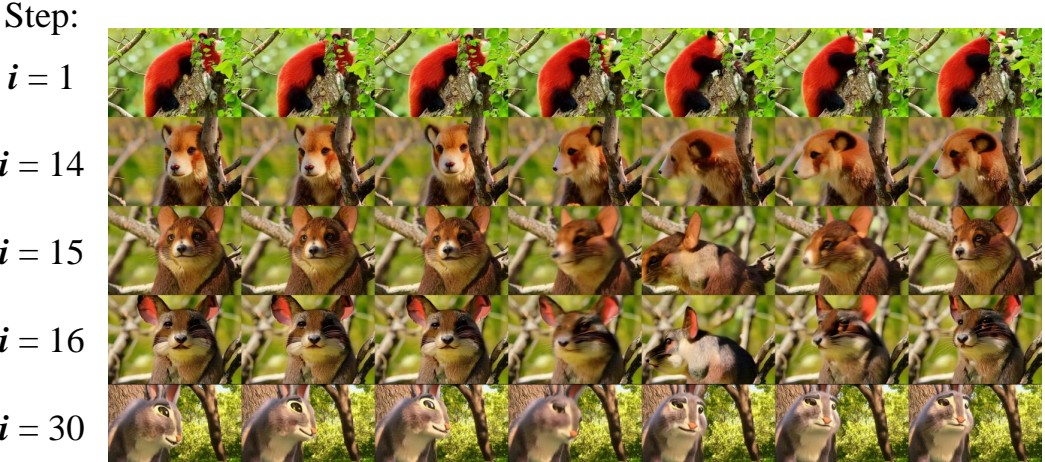

Prompt B: *"The **rabbit** sat still under a tree."*

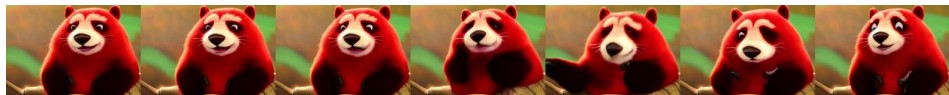

Prompt C: *"The **red panda** with a rabbit face, rested on a sturdy branch."*

Figure 34: **Mixture of ["Red Panda"] and ["Rabbit"]**. Our objective is to mix the features described in Prompt A and Prompt B with the guidance of Prompt C. We set the total number of interpolation steps to 30. Using Algorithm 1, we identify the 15-th interpolation embedding as the optimal embedding and generate the corresponding video. The video generated directly from Prompt C does not exhibit the desired mixed features from Prompts A and B.

**Algorithm 4** 3D Attention

---

1: **datastructure** 3D ATTENTION         ▷ Definition 4.7
2: **members**
3:     $n \in \mathcal{N}$: the length of input sequence
4:     $n_f \in \mathcal{N}$: the number of frames
5:     $h \in \mathcal{N}$: the hight of video
6:     $w \in \mathcal{N}$: the width of video
7:     $d \in \mathcal{N}$: the hidden dimension
8:     $c \in \mathcal{N}$: the channel of video
9:     $c_{\text{patch}} \in \mathbb{R}^{n \times d}$: the channel of patch embedding.
10:    $E_t \in \mathbb{R}^{n \times d}$: the text embedding.
11:    $E_{\text{video}} \in \mathbb{R}^{n_f \times h \times w \times c}$: the video embedding.
12:    $E_{\text{patch}} \in \mathbb{R}^{n_f \times h' \times w' \times c_{\text{patch}}}$: the patch embedding.
13:    $\phi_{\text{conv}}(X, c_{\text{in}}, c_{\text{out}}, p, s)$: the convolution layer.     ▷ Definition 4.5
14:    $\text{Attn}(X)$: the attention block.     ▷ Definition 4.4
15:    $\phi_{\text{linear}}(X)$: the linear projection.     ▷ Definition 4.6
16: **end members**
17:
18: **procedure** 3D ATTENTION($E_t \in \mathbb{R}^{n \times d}, E_v \in \mathbb{R}^{n_f \times h \times w \times c}$)
19:    /* $E_{\text{patch}}$ dimension: $[n_f, h, w, c_v] \to [n_f, h', w', c_{\text{patch}}]$ */
20:    $E_{\text{patch}} \leftarrow \phi_{\text{conv}}(E_v, c_v, c_{\text{patch}}, p = 2, s = 2)$
21:    /* $E_{\text{patch}}$ dimension: $[n_f, h', w', c_{\text{patch}}] \to [n_f \times h' \times w', c_{\text{patch}}]$ */
22:    $E_{\text{patch}} \leftarrow \text{reshape}(E_{\text{patch}})$
23:    /* $E_{\text{hidden}}$ dimension: $[n + n_f \times h' \times w', c_{\text{patch}}]$ */
24:    $E_{\text{hidden}} \leftarrow \text{concat}(E_t, E_{\text{patch}})$
25:    /* $E_{\text{hidden}}$ dimension: $[n + n_f \times h' \times w', c_{\text{patch}}]$ */
26:    $E_{\text{hidden}} \leftarrow \text{Attn}(E_{\text{hidden}})$
27:    /* $E_t$ dimension: $[n, d]$ */
28:    /* $E_{\text{patch}}$ dimension: $[n_f \times h' \times w', c_{\text{patch}}]$ */
29:    $E_t, E_{\text{patch}} \leftarrow \text{split}(E_{\text{hidden}})$
30:    /* $E_v$ dimension: $[n_f \times h' \times w', c_{\text{patch}}] \to [n_f \times h \times w, c_v]$ */
31:    $E_v \leftarrow \phi_{\text{linear}}(E_{\text{patch}})$
32:    /* $E_v$ dimension: $[n_f \times h \times w, c_v] \to [n_f, h, w, c_v]$ */
33:    $E_v \leftarrow \text{reshape}(E_v)$
34:    Return $E_v$
35: **end procedure**

---

## LLM USAGE DISCLOSURE

LLMs were used only to polish language, such as grammar and wording. These models did not contribute to idea creation or writing, and the authors take full responsibility for this paper's content.

