# OpenReview forum: "RichSpace: Enriching Text-to-Video Prompt Space via Text Embedding Interpolation"
_ICLR.cc/2026/Conference — ICLR 2026 Conference Withdrawn Submission_

### Official Review · Reviewer_uoUC · 2025-10-27

**Soundness:** 2
**Presentation:** 1
**Contribution:** 2
**Rating:** 2
**Confidence:** 4

**Summary:**

The authors proposed an optimal embedding interpolation approach between multiple prompts via leveraging perpendicular foot embeddings and cosine similarity, thereby enabling text-to-video models to better follow unusual prompts, e.g., “a tiger with zebra-like
stripes walking on grassland”.

After carefully reviewing the submitted manuscript, I find that the paper’s novelty is quite limited, the proposed approach is overly simplistic and lacks sufficient technical depth, and the overall presentation is poorly structured. These issues significantly lower the contribution level, making it fall short of the standards expected for a top-tier venue like ICLR.

**Strengths:**

1. The proposed method is training-free and can be applied to existing text-to-video generation models without architecture changes.

**Weaknesses:**

1. It seems that the paper's novelty is very limited. The paper does not introduce fundamentally new ideas or insights, and the overall contribution appears incremental.

2. The proposed approach is overly simplistic and lacks substantial technical depth, just prompt embeddings interpolation, without offering significant algorithmic or theoretical innovation for subsequenct research.

3. The paper is not well structured, which affects readability and understanding. For example, key methodological details are not clearly presented in Section 2.2, and many essential definitions are deferred to Section 4.2, making the method difficult to follow. The authors should comprehensively polish the manuscript to make logical exposition clearer and more coherent.

4. The experiments are not solid. For example, the results in Table 1 are very inconvincing since the authors just list the results on some carefully selected examples. The authors should report the performance averaged over 100 or 1000 samples, which will better support the argument provided in this paper.

5. The manuscript may include several typos; the authors should carefuly revised them, for example, the line 10 of Algorithm 2 and the second cell of the first column of Table 1, which may confuse the paper.

6. Only subject consistency and aesthetic quality are reported; more metrics (e.g., FVD, CLIPSIM, temporal consistency) would strengthen claims.

7. I recommend the authors to conduct user study, which may better validate the effectiveness of the propose method.

8. I still question the value of considering such unusual and unrealistic scenarios, for example, “a red panda with a rabbit face” or “an elephant with the face of a lion.” These scenarios seem largely meaningless and may not provide practical or insightful contributions to real-world applications.

As I really believe that the novelty of this paper is largely lower than the bar of ICLR; so, my current decision is rejection but may chage during author rebuttal.

**Questions:**

See weaknesses

---

> ### Author Response · Authors · 2025-12-01
>
> Thank you for your thoughtful feedback. Your comments are very helpful and much appreciated. We will address these in the next version.

---

### Official Review · Reviewer_zXaP · 2025-10-28

**Soundness:** 3
**Presentation:** 3
**Contribution:** 3
**Rating:** 6
**Confidence:** 4

**Summary:**

This paper demonstrates that selecting the right text embedding can effectively guide video generation models to produce desired videos. It also proposes a simple yet effective algorithm that uses vertical projection embedding and cosine similarity to find the optimal text embedding.

**Strengths:**

The method described in the paper is simple yet effective. By employing certain embedding design techniques, it can enhance the performance of video foundation models, which is somewhat novel. The descriptions of the images and experiments are also clear. It would be meaningful if the design in the textual part could further improve the performance of video foundation models, potentially avoiding the need for fine-tuning most parameters in the backbone.

**Weaknesses:**

A major confusion is whether this method can improve performance in text-to-image generation? If it can, why haven't related experiments been conducted? If not, what are the factors that hinder such improvements? What is the relationship between different text embeddings and temporal consistency?

In the experiment phase of this paper, it would be more convincing if some dynamic videos could be shown. I am curious whether this method might affect the temporal consistency of videos, or whether there would be any video jittering phenomena.

Moreover, among the examples presented, there are fusions of two objects that have never been seen before. I understand that in the training of video generation models, it is rare for non-natural scenes or synthetic videos to be included in the training data (perhaps synthetic images are more commonly used in image model training).

The examples shown in Table 1 are too few. Should we build a dataset to broadly demonstrate the effectiveness of the proposed method?

The experimental section is too brief in the main text. Algorithms 1 and 2 take up a significant portion of the text, which seems unreasonable. The figures in the experimental section are too small.

**Questions:**

see Weaknesses

---

> ### Author Response · Authors · 2025-12-01
>
> Thank you for your thoughtful feedback. Your comments are very helpful and much appreciated. We will address these in the next version.

---

### Official Review · Reviewer_XZWH · 2025-10-31

**Soundness:** 3
**Presentation:** 2
**Contribution:** 2
**Rating:** 2
**Confidence:** 4

**Summary:**

This paper proposes RichSpace, a training-free method to improve text-to-video (T2V) generation on compositional prompts (e.g., “tiger with zebra stripes”). The key idea is to linearly interpolate between the embeddings of two base prompts (A, B), then select the best interpolation step via a perpendicular-foot projection of a guidance prompt (C) and cosine similarity, and finally generate the video using that interpolated embedding. The paper argues that text encoders are a practical bottleneck and offers a formal result that finite text spaces cannot cover the continuous video space. Qualitative examples and a small quantitative study (VBench Subject Consistency/Aesthetic Quality) on CogVideoX-2B are provided.

**Strengths:**

- Simple, plug-and-play, training-free: No model changes; just embed A/B/C, scan k linear interpolants between A and B, rank with a projection-based cosine score, and generate with the best step. Clear, reproducible algorithms (Alg. 1–3).

- Padding-aware scoring detail: Computes similarity both on full (padded) and truncated embeddings, then combines them—acknowledges real encoder quirks.

- Addresses a real pain point (compositionality): Many qualitative cases where direct prompting with C fails to blend, but the chosen interpolant succeeds; also demonstrates three-prompt chaining.

- Some quantitative signal: Slight improvement in Subject Consistency on VBench; authors transparently report an AQ trade-off.

**Weaknesses:**

- Limited novelty: Core method (linear interpolation + perpendicular-foot anchor + cosine ranking) is a neat heuristic but conceptually close to existing latent/prompt search ideas; largely a prompt-space tweak.

- Methodological narrowness: Search is constrained to the straight line between embeddings on a highly non-linear manifold; token-level structure is mostly ignored (mean cosine over rows).

- Theory–practice mismatch: The theorem shows global coverage limits of text→video mappings, but experiments target encoder mis-embedding of an existing textual description C; the proof neither diagnoses this failure nor justifies why linear interpolation + cosine to a projection should fix it.

- Narrow evaluation: Only CogVideoX-2B, mostly toy compositional prompts (animals/objects); no broader instruction following, temporal reasoning, or camera control; no human study.

- No strong baselines: Missing comparisons to CLIP-guided latent search, gradient-based prompt tuning, token-level composition controllers, or multi-encoder fusion approaches. (Not reported.)

- Modest quantitative lift with trade-off: Subject Consistency improves slightly (≈0.9748→0.9787) but Aesthetic Quality drops (≈0.5519→0.5163); interpretation that lower AQ reflects “rarer desired blends” is plausible but unvalidated without human ratings.

**Questions:**

- Token awareness: Can you weight or select tokens (e.g., nouns vs. attributes) during interpolation/selection instead of averaging row-wise cosine? Any analysis of attention maps or token salience?
- Selection metric: Why is cosine to the perpendicular foot the right proxy? Have you tested alternative selection criteria (e.g., CLIP-text similarity to a canonical mixture caption, cross-attention probes, learned verifiers)?

---

> ### Author Response · Authors · 2025-12-01
>
> Thank you for your thoughtful feedback. Your comments are very helpful and much appreciated. We will address these in the next version.

---

### Official Review · Reviewer_o8gy · 2025-11-04

**Soundness:** 2
**Presentation:** 2
**Contribution:** 2
**Rating:** 4
**Confidence:** 4

**Summary:**

This paper addresses the common failure of text-to-video (T2V) models to generate videos for complex, compositional prompts (e.g., "a tiger with zebra-like stripes"). The authors' contributions are twofold: 1) A theoretical proof (Theorem 5.9) arguing that the discrete prompt space is insufficient to map to the entire continuous video space, thus identifying a "gap". 2) A training-free algorithm (Algorithm 1) proposed as a solution. This algorithm attempts to find an "optimal" embedding in this gap by first linearly interpolating between two base embeddings (e.g., "tiger" and "zebra"). It then uses a third "guide" prompt ("tiger with stripes") to find a "perpendicular foot" anchor and selects the best interpolation step via a complex cosine similarity metric.

**Strengths:**

Theoretical Motivation: Section 5 provides a formal proof that a "gap" exists between the discrete prompt space and the continuous video space, given a model $f$. This motivates manipulation on embedding space.

Simple Algorithm: This paper showed that simple linear interpolation can produce results that are more aligned than using the target text prompt directly.

**Weaknesses:**

Weak theoretical analysis: The paper provides a formal proof demonstrating that text embeddings cannot represent the full spectrum of all possible videos. While this conclusion is reasonable, it is also somewhat trivial. This finding, however, does not directly imply that desired results are unattainable using counter-intuitive or out-of-distribution text prompts, nor does it preclude the possibility that more nuanced prompting could address this limitation.

Weak Quantitative Evidence: The experimental results are weak. The quantitative gains in "Subject Consistency" are marginal (0.9787 vs 0.9748). The authors' defense of the lower "Aesthetic Quality" score could be unconvincing speculation. It could be that these out-of-distribution interpolated embeddings simply produce lower-fidelity, artifact-filled videos.

No baselines and ablations: Although a simple method is proposed, it lacks an ablation study to clearly understand the individual contribution of each component. For example, the direct effects of CosTruc and CosFull are unclear, nor is it certain that simply adding the corresponding embeddings is the optimal approach.

Weak presentation: While the overall logical structure the paper tries to explain makes sense, the explanation is overly verbose given the simplicity of the algorithm. It would have been much easier to grasp the concept if there had been a figure visually explaining the core interpolation. Furthermore, the paper appears unpolished in general, for instance, by referencing a non-existent figure in the sentence: "Furthermore, in Figure2 (c),we observe the same behavior in the domain of plants, specifically with the combination of rose and cactus features."

**Questions:**

(Re: Weakness 1) Can you provide any justification, theoretical or empirical, for the core assumption that the optimal embedding for a complex new concept lies on the 1D linear interpolation path between two base embeddings? Why would this be true in a high-dimensional, non-linear space?

(Re: Weakness 2) Do you have any evidence that the lower Aesthetic Quality score is due to "novelty" and not simply due to the generation of videos with more artifacts?

(Re: Weakness 3)  A detailed ablation study is needed. This should include an analysis of the specific benefits of using CosTruc and CosFull, as well as comparisons against simpler baselines such as naive interpolation (e.g., a 50/50 split) and spherical interpolation..etc.

(Re: Weakness 3) How sensitive is your algorithm to the phrasing of the guide prompt $P_c$? This algorithm relies heavily on the careful selection of embeddings A and B, which serve as anchors. For this methodology to be considered generally useful, a standard for this process seems necessary. The prompts used in the paper appear inconsistent; some seem rule-based, while others do not. Is there a standard format for defining the anchor instructions A and B, based on a reasonable set of words to 'fix' and words to 'merge'? Furthermore, to better validate the method's effectiveness, could such a format be applied to a much larger dataset instead of relying on manual, case-by-case prompting? The current experimental data is too scarce, making it difficult to determine if this is a generally effective methodology or one that merely works well on these few hand-picked samples.

---

> ### Author Response · Authors · 2025-12-01
>
> Thank you for your thoughtful feedback. Your comments are very helpful and much appreciated. We will address these in the next version.

---

### Note · Authors · 2025-12-01

I have read and agree with the venue's withdrawal policy on behalf of myself and my co-authors.